# AlphaRouter: Token-level Routing Between SLM and LLM with Reinforcement Learning and Tree Search

Siteng Liao [1 2]   Yuzhu Liang [2]   Hengzhong Rao [1 2]   Xizhao Luo [3]   Tian Wang [2]

## Abstract

SLM–LLM routing accelerates generation by strategically invoking LLMs for critical tokens. However, existing methods typically train routers to mimic the LLM, capping performance at the reference trajectory's limit. In this work, we demonstrate that the SLM–LLM collaborative inference space offers a richer solution set, yielding correct answers even when the LLM fails. To exploit this, we propose **AlphaRouter**, a routing framework learning optimal collaborative inference paths via a *search and iterative update* paradigm. Formulating routing as a Markov Decision Process, we introduce Collaborative Inference Tree Search (CITS) to explore the solution space. To tackle the severe credit assignment challenge posed by sparse rewards, we propose Tree-Advantage Policy Optimization (TAPO) to optimize the routing policy. By leveraging counterfactual advantages within the tree structure, TAPO effectively attributes the final reward to specific routing decisions, stabilizing training without dense supervision. Extensive experiments show that AlphaRouter advances the Pareto frontier of accuracy-efficiency trade-offs by exploiting better inference trajectories in the collaborative space. Code is available at https://github.com/StripeLife0217/AlphaRouter.

## 1. Introduction

In recent years, by scaling up model parameters and training data, Large Language Models (LLMs) have achieved remarkable performance on complex tasks (Achiam et al.,

[1]Faculty of Arts and Sciences, Beijing Normal University, Zhuhai, China [2]BNU Institute of Artificial Intelligence and Future Networks, Beijing Normal University, Zhuhai, China [3]School of Computer Science and Technology, Suzhou University, Suzhou, China. Correspondence to: Yuzhu Liang <yuzhuliang@bnu.edu.cn>.

*Proceedings of the 43rd International Conference on Machine Learning*, Seoul, South Korea. PMLR 306, 2026. Copyright 2026 by the author(s).

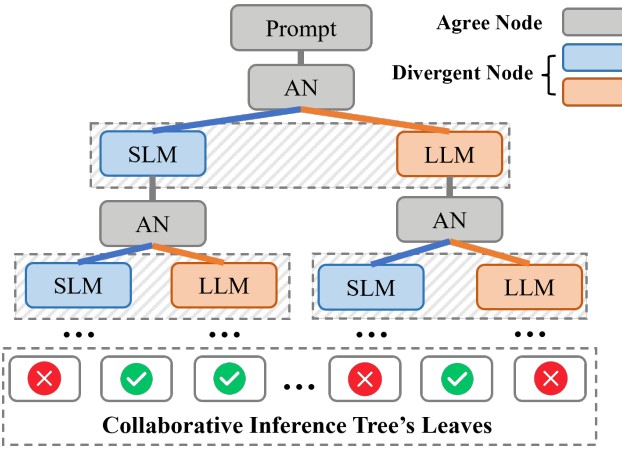

*Figure 1.* **Collaborative Inference Tree Search (CITS).** Unlike chain-based sampling, CITS explores the expanded collaborative strategy space by branching at Divergent Nodes.

2023; Liu et al., 2024; Team et al., 2023; Kaplan et al., 2020). However, the substantial computational overhead inherent in these massive models leads to high latency and prohibitive inference costs, restricting their deployment in latency-sensitive applications (Qu et al., 2025; Chen et al., 2025; Jiang et al., 2024). In contrast, Small Language Models (SLMs) offer significantly faster inference speeds due to their compact architectures but often suffer from limited capabilities (Kamalloo et al., 2024). To bridge this gap, *SLM–LLM routing* aims to design strategies that maintain near-LLM accuracy while minimizing expensive LLM invocations (Sun et al., 2024; Yue et al., 2025).

Existing SLM–LLM routing methods can be broadly categorized into two levels: *query-level routing (QR)* (Chen et al., 2023; Ding et al., 2025; Chuang et al., 2025; Sikeridis et al., 2025; Ong et al., 2025; Feng et al., 2025; Lu et al., 2024) and *token-level routing (TR)* (Shen et al., 2024; Zheng et al., 2025; Fu et al., 2025). Query-level routing dispatches an input query to a single model based on query-level representations. While QR is highly efficient, its granularity is coarse: once a query is routed to the LLM, all tokens are generated by the LLM. Yet not all tokens require the capability of a large model; many can be safely produced by an SLM, further reducing latency without sacrificing accuracy.

This limitation motivates token-level routing, which makes a routing decision at each generation step. Typically, TR defaults to generating the next token with the SLM, and uses the contextual representations and statistics derived from the SLM to decide whether the LLM should generate the next token. This fine-grained strategy enables the router to invoke the LLM only at critical positions (Varangot-Reille et al., 2025).

However, the recent TR methods mostly train the router by aligning the collaborative inference trajectory to the LLM's own inference path, which inherently caps the upper bound of collaboration by the LLM's inference accuracy (Shen et al., 2024; Fu et al., 2025). Although intuitive, we observe that for the vast majority of cases where the LLM performs poorly, there exist better trajectories in the collaborative inference space. To validate this, building on the observation in R2R (Fu et al., 2025) that SLM and LLM tokens under the same prefix often coincide, we explore counterfactual inference trajectories via a budgeted tree search. Concretely, starting from the root, when the SLM and LLM produce the same token under an identical prefix, we expand an agree node (AN) with a single child; only when they differ do we expand a diverge node (DN) with two children, forming two subtrees that represent trajectories induced by different next tokens. On 1,000 GSM8K test instances (Cobbe et al., 2021), using a 0.5B SLM and a 7B LLM, this procedure finds a correct-answer trajectory for up to 99% of the problems (while the 7B LLM achieves 90.2% accuracy on the same set). This substantial gap suggests that optimal routing is not merely about mimicking the expert, but about discovery. This observation leads to a central research question: *Is it possible to learn a routing policy that approximates these optimal collaborative trajectories without expensive test-time search?*

Answering this question faces three primary challenges. **(1) Exponential Search Space.** The collaborative tree grows as $\mathcal{O}(2^n)$, rendering exhaustive search intractable and necessitating a budget-constrained, iterative optimization mechanism. **(2) Sequential Dependency.** Due to autoregressive generation (Vaswani et al., 2017), current token choices shift future distributions. This frames routing as a Markovian *sequential decision-making* problem rather than independent binary classification, making reinforcement learning a natural fit. **(3) Credit Assignment with Sparse Rewards.** Relying solely on final answer correctness yields extremely sparse signals. Attributing the final outcome to specific intermediate routing decisions remains a severe challenge, often leading to unstable optimization (Pignatelli et al., 2024).

To address these challenges, we propose **AlphaRouter**, a reinforcement learning-based token-level routing framework inspired by the *search and iterative update* philosophy (Silver et al., 2018). We formalize collaborative inference as a Markov Decision Process (MDP) and introduce **Collaborative Inference Tree Search (CITS)** to efficiently explore the solution space. Unlike linear sampling, CITS constructs a hierarchical structure that aggregates consensus at AN while explicitly expanding counterfactual branches at DN. To resolve the credit assignment difficulty under sparse rewards, we propose **Tree-Advantage Policy Optimization (TAPO)**. Instead of relying on noisy global rewards, TAPO computes the *relative advantage* of a routing action by contrasting the expected outcomes of the sibling subtrees at each divergence point. This mechanism effectively isolates the impact of specific routing decisions, stabilizing the learning process. Our contributions are summarized as follows:

- **New Paradigm:** We propose AlphaRouter, which shifts the training paradigm of token-level routing from mimicking a static LLM to discovering optimal trajectories via *search and iterative updates*, effectively unlocking the latent potential of the collaborative solution space beyond the LLM's default trajectory

- **Core Algorithm:** We introduce the CITS mechanism to construct a compact collaborative state space and the TAPO algorithm to solve the credit assignment problem. TAPO leverages the tree structure to compute fine-grained advantage signals from counterfactual subtrees, ensuring robust optimization under sparse supervision.

- **SOTA Performance:** Extensive experiments demonstrate that AlphaRouter pushes the Pareto frontier of accuracy versus inference cost. It significantly outperforms state-of-the-art baselines and, in certain configurations, achieves higher accuracy than the standalone LLM by exploiting superior inference paths within the collaborative space.

## 2. Related Works

### 2.1. Query-level Routing

Query-level routing treats model selection as a coarse-grained decision, aiming to process an entire input query with the most appropriate model configuration (Huang et al., 2025). A foundational approach in this category is the LLM cascade strategy proposed in FrugalGPT (Chen et al., 2023), which sequentially invokes models from cheapest to most expensive, stopping early if a lightweight model produces a sufficiently high-quality response. Moving beyond sequential cascades to direct selection, RouteLLM (Ong et al., 2025) formulates routing as a recommendation problem, utilizing matrix factorization and causal LLM classifiers trained on human preference data to predict model win-rates. To better capture the complex interactions between tasks and models, GraphRouter (Feng et al., 2025) constructs a heterogeneous graph and employs graph neural networks

to generalize routing decisions to unseen LLMs. Beyond simple accuracy, practical deployment requires optimizing multiple objectives. PickLLM (Sikeridis et al., 2025) adopts a reinforcement learning framework with a weighted reward function to dynamically balance cost, latency, and response quality. Similarly, ZOOTER (Lu et al., 2024) leverages reward models as silver supervision to distill efficient routing policies. Recent studies have also explored leveraging internal model uncertainty (Ma et al., 2025); for instance, (Chuang et al., 2025) introduce "confidence tokens" to determine whether a model should answer or fallback to an expert. Furthermore, BEST-Route (Ding et al., 2025) argues that simple routing is insufficient for difficult queries and proposes an adaptive test-time compute strategy that dynamically adjusts the number of samples based on query difficulty. Despite their effectiveness in reducing average costs, query-level approaches suffer from an "all-or-nothing" limitation, often failing to address cases where a SLM falters only at specific, critical steps of a long generation.

## 2.2. Token-level Routing

Token-level routing, or collaborative decoding, addresses the granularity limitation by enabling dynamic model switching during the generation process. This paradigm aims to route only "critical" or "difficult" tokens to the LLM while offloading the majority of the generation to a SLM. Co-LLM (Shen et al., 2024) models the routing decision as a latent variable, allowing a base model to learn when to defer generation to an assistant model in an unsupervised manner. To explicitly optimize for efficiency and quality, CITER (Zheng et al., 2025) formalizes the router training as a policy optimization problem, learning to predict token-level routing scores that distinguish between critical and non-critical tokens based on future rewards. Most recently, R2R (Fu et al., 2025) provides a deeper insight into the collaboration mechanism by analyzing inference paths. It observes that SLMs and LLMs agree on the vast majority of tokens, with differences often being semantically neutral. Consequently, R2R proposes to trigger the LLM only when a genuine divergence in inference paths is detected, thereby minimizing overhead without compromising the inference capabilities of the LLM. These fine-grained approaches represent the state-of-the-art in efficient inference, allowing for precise resource allocation within a single generation sequence.

## 3. Token-level Routing Problem

We consider the task of token-level collaborative inference, involving a LLM, denote as $\pi_l$, and a SLM, $\pi_s$. Given an input prompt $x$, the system generates a target sequence $Y = (y_1, y_2, \ldots, y_T)$. At each generation step $t$, a router policy $\pi_r(a_t|x, y_{<t})$ outputs a probability distri-

bution over the binary action space $a_t \in \{s, l\}$. The next token $y_t$ is then sampled from the selected model. Similar to (Zheng et al., 2025; Fu et al., 2025), we focus on greedy sampling for simplicity, which can be denoted as $y_t \sim \arg\max_y \pi_{a_t}(y|x, y_{<t})$.

The goal of token-level routing is to maximize the generation quality while minimizing the computational overhead. We formalize this as optimizing the policy $\pi_r$ to maximize the expected return over the trajectory $\tau$:

$$\max_{\pi_r} \mathcal{J}(\theta) = \mathbb{E}_{\tau \sim \pi_r} \left[ V(x, Y) - \lambda \sum_{t=1}^{T} \mathbb{1}(a_t = l) \right]. \quad (1)$$

Here, the expectation is taken over the trajectory $\tau$ induced by sampling actions and tokens. $V(x, Y)$ represents the quality of the final output, and the second term imposes a penalty for each LLM invocation to encourage efficiency.

## 4. Method

In this section, we introduce **AlphaRouter**, a framework designed to learn optimal collaborative inference trajectories via search and iterative updates. We begin by formulating the token-level routing process in Section 3 as a *Markov Decision Process (MDP)*. Subsequently, we apply a hierarchical probabilistic decomposition to the routing policy $\pi_r$, factorizing it into two components: *DN Identification* and *DN Routing*. These components are responsible for identifying DNs and making routing decisions at these nodes, respectively. Building on this foundation, we propose a budget-constrained **Collaborative Inference Tree Search (CITS)**. This method efficiently collects tree-structured empirical samples within an acceptable training budget. By generating highly correlated trajectories that branch at DNs, CITS provides high-quality supervision for training the DN Routing policy. Finally, we introduce a two-stage training framework to optimize these decomposed policies. We first employ our proposed **Tree-Advantage Policy Optimization (TAPO)** to train DN Routing policy, and subsequently train the DN Identification module, ensuring efficient inference and strict budget adherence.

### 4.1. MDP Reformulation

To optimize the objective function $\mathcal{J}(\theta)$ via reinforcement learning, we reformulate the token-level routing process as a Markov Decision Process (MDP) (Sutton et al., 1998). In this framework, the fixed SLM and LLM act as the stochastic environment, and the router acts as the agent. We define the key elements of the MDP as follows:

**State Space $\mathcal{S}$:** Similar to (Fu et al., 2025; Wang et al., 2025), we construct a dense state representation $s_t$ to capture

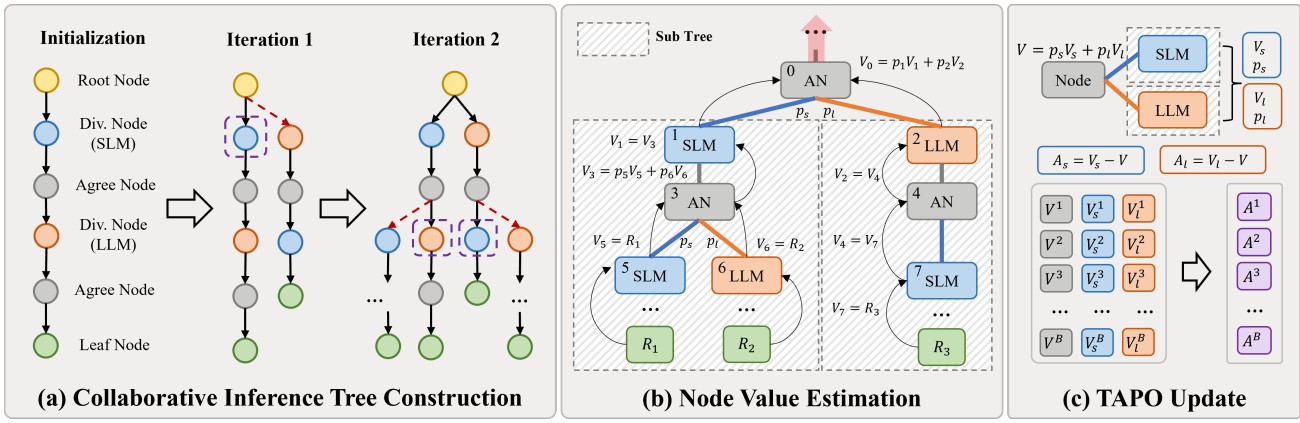

**(a) Collaborative Inference Tree Construction**   **(b) Node Value Estimation**   **(c) TAPO Update**

*Figure 2.* **The overall framework of AlphaRouter.** (a) **Collaborative Inference Tree Construction**: CITS identifies DNs from trajectories and iteratively expands the search tree by exploring counterfactual actions to build a diverse set of counterfactual trajectories. (b) **Node Value Estimation**: The value of each node ($V$) is recursively estimated as the expected future reward from its sub-trees, where ANs serve as deterministic transitions and DNs aggregate rewards via probabilistic action weighting. (c) **TAPO Update**: TAPO calculates the advantage ($A$) of SLM and LLM actions relative to the node's expected value, transforming the insights from tree-search into a high-performance, lightweight routing policy via reinforcement learning.

the model's uncertainty and generation context. Specifically, given the input $x$ and prefix $y_{<t}$ fed into the SLM, the state $s_t$ is composed of: $s_t = [\mathbf{h}_t, \mathbf{l}_t, \mathbf{e}_t]$, where $\mathbf{h}_t^{\text{last}}$ denotes the SLM's last hidden layer state, $\mathbf{l}_t^{\text{top-}k}$ represents the top-$k$ logits indicating the SLM's confidence, and $\mathbf{e}_{t-1}$ is the embedding of the generated token.

**Action Space $\mathcal{A}$:** The action space is binary, $\mathcal{A} = \{s, l\}$. Action $a_t = s$ selects the SLM $\pi_s$, while $a_t = l$ selects the LLM $\pi_l$.

**Transition Dynamics $\mathcal{P}$:** The transition is driven by the token sampling of the selected model. Given $a_t$, the next token is sampled as $y_t \sim \pi_{a_t}(\cdot|x, y_{<t})$. The state then updates to $s_{t+1}$ based on the new prefix.

**Reward Function $\mathcal{R}$:** We decompose the trajectory-level objective into step-wise rewards $r_t$ to facilitate temporal credit assignment. The reward at each step $t$ is defined as:

$$r_t = \begin{cases} -\lambda \mathbb{1}(a_t = l) & \text{if } t < T, \\ V(x, Y) - \lambda \mathbb{1}(a_t = l) & \text{if } t = T. \end{cases} \quad (2)$$

Here, intermediate steps only incur the latency penalty $\lambda$ if the LLM is used. The final step $T$ incorporates the generation quality reward $V(Y)$. The cumulative return of a trajectory, $R(\tau) = \sum_{t=1}^{T} r_t$, remains consistent with the global objective $\mathcal{J}(\theta)$.

### 4.2. Policy Decomposition

In this section, we align the definition of tree nodes with the MDP formalism. However, the action space for the collaborative policy scales to $O(2^n)$, rendering exhaustive exploration at every token step computationally infeasible. Leveraging the statistical observation that SLMs and LLMs

share a high probability of outputting identical tokens given the same prefix, we restrict exploration effectively to instances where the models diverge.

For ANs, the current decision is independent of future token routing. Therefore, to minimize computational costs, the SLM should be selected deterministically. Consequently, the routing decision on ANs is formulated as a binary classification task. Conversely, for DNs, tokens generated by the SLM and LLM may lead to distinct inference trajectories, influencing subsequent routing decisions and ultimately yielding different inference outcomes. We therefore treat the routing problem for DNs as a sequential decision-making task.

To efficiently solve this, we structure the global routing policy as a *hierarchical probabilistic decomposition*. The policy is parameterized by two independent networks: a DN identifier $\pi_{node}$ and a DN Router $\pi_{div}$. We define $\pi_{node}(s)$ as the probability scalar classifying the current node as a DN. Furthermore, we define $\pi_{div}(s)$ as the conditional probability of utilizing the LLM, given that the node is identified as a DN. Formally, we express the global policy $\pi_r(s)$ as the product of the node classification probability and the conditional routing probability:

$$\pi_r(s) \triangleq P(a = l|s) = \underbrace{\pi_{node}(s)}_{P(\text{DN}|s)} \cdot \underbrace{\pi_{div}(s)}_{P(a=l|\text{DN},s)} . \quad (3)$$

### 4.3. Collaborative Inference Tree Search

To effectively explore the inference space during the training phase under limited computational resources, we propose budget-constrained **Collaborative Inference Tree**

**Search(CITS)**. This algorithm constructs the inference tree $\mathcal{T}$ incrementally over $N$ rounds, dynamically allocating the computational budget to generate a final set of $M$ inference trajectories.

We define the total path budget as $M$ and the maximum expansion rounds as $N$. The core of our approach lies in the *synchronous inference* mechanism, where both the SLM and LLM generate tokens given the same prefix. The algorithm proceeds in $N$ iterative rounds ($1 \leq t \leq N$), maintaining a global candidate set $\mathcal{C}$ of unexpanded divergent nodes. The execution flow is structured as follows:

**Collaborative Path Extension.** In each round, we extend the active leaf nodes of the current tree. For a given prefix, both models output their predicted next tokens, $y_s$ and $y_l$. If the tokens are identical ($y_s = y_l$), we classify the current state as an AN and append the token to the prefix, and the extension continues automatically to the next timestep. If the tokens differ ($y_s \neq y_l$), the current node is identified as a DN. We add this node to the global expansion candidate set $\mathcal{C}$. To continue the current path, we sample the next token based on the current divergent router $\pi_{div}$ and proceed with the generation.

**Dynamic Budget Allocation.** At the beginning of iteration $t$, we determine how many new branches to spawn. Let $|\mathcal{P}_{curr}|$ denote the number of completed paths in $\mathcal{T}$. We calculate the remaining path $M_{remain} = M - |\mathcal{P}_{curr}|$. To ensure exploration stability across all $N$ rounds, the expansion budget $K_t$ for the current round is dynamically calculated as:

$$K_t = \left\lceil \frac{M - |\mathcal{P}_{curr}|}{N - t + 1} \right\rceil. \qquad (4)$$

This formula ensures uniform budget consumption. If $M_{remain} \leq 0$, the expansion process terminates.

**Candidate Sampling & Counterfactual Expansion.** Once $K_t$ is determined, we select expansion points from the candidate set $\mathcal{C}$ to explore counterfactual inference paths. We randomly sample $\min(K_t, |\mathcal{C}|)$ nodes from $\mathcal{C}$. For each selected node $v$, we perform a **counterfactual expansion**: We identify the action $a$ taken in the existing path. We force the new branch to take the counterfactual action $\neg a$. From this new state, we initiate a new *Collaborative Path Extension* process (Phase 1), allowing the algorithm to hierarchically refine the tree structure from the backbone to the details. The pseudocode of CITS is provided in Appendix A.1.

**Training-only Search.** CITS is used only as an offline training procedure. During deployment, AlphaRouter does not expand a tree, maintain multiple active paths, or select from multiple generated answers. The trained node identifier and DN router instead execute a single autoregressive routed trajectory, so test-time evaluation follows the same single-output protocol as the standalone baselines. In par-

ticular, HumanEval is evaluated as true pass@1 rather than pass@$M$.

**4.4. Tree-Advantage Policy Optimization**

Based on the inference trajectories organized within the CIT structure, and inspired by the Tree-GRPO (Ji et al., 2025), we propose the Tree-Advantage Policy Optimization (TAPO) tailored for the DN router. Designed to address the challenge of credit assignment in complex inference chains, TAPO leverages the global perspective of the CIT to iteratively update the routing policy. The optimization procedure operates in two consecutive stages: first, a bottom-up tree-backup value estimation to approximate the long-term expected return of divergent states; and second, a policy update mechanism utilizing relative advantage to ensure stable convergence towards the optimal collaborative inference strategy.

4.4.1. TREE-BACKUP VALUE ESTIMATION

To optimize the routing policy for DNs, we need to approximately assess the long-term value $V(s)$ of each node. TAPO leverages the CITS tree structure to backpropagate value bottom-up via Post-order Traversal.

For nodes with only a single successor (including all ANs, as well as unexpanded DNs where only a single path was explored), the value depends entirely on the subsequent performance of that unique path. Therefore, the parent node directly inherits the value of its child $s'$:

$$V(s) = V(s'), \quad \text{if } |\text{children}(s)| = 1. \qquad (5)$$

For DNs that are expanded, the value of the parent node is defined as the weighted expectation of the values of the two branches under the current policy $\pi_{old}$. Let $s_s$ and $s_l$ be the child nodes resulting from the corresponding actions; then:

$$V(s) = \pi_{old}(a_s|s) \cdot V(s_s) + \pi_{old}(a_l|s) \cdot V(s_l), \qquad (6)$$

where $\pi_{old}$ represents the policy probability when sampling. Through this tree-backup mechanism (Daley & Chan, 2022), $V(s)$ captures both the precise feedback from deterministic paths and the smooth estimated expected return from divergent paths, providing a low-variance value baseline for subsequent policy updates. Furthermore, leveraging the CIT organization, the value of a divergent node is derived exclusively from inference paths sharing the same prefix. This design not only facilitates natural credit attribution but also significantly enhances sample efficiency.

4.4.2. POLICY UPDATE WITH ADAPTIVE ENTROPY

Upon obtaining robust node value estimates via CIT construction and Tree-backup estimation, we proceed to the second phase: policy optimization for DNs. Unlike traditional

single-path policy gradient methods that suffer from high variance due to the lack of baseline comparisons, TAPO leverages the structural advantage of the CIT to perform more precise policy updates at DNs.

Specifically, for each expanded DN state $s$ in the sampled trajectory and its corresponding action $a$, we compute the advantage value $A(s, a)$. The key innovation is that our value estimate $V(s)$ aggregates information from both the executed path and the explored counterfactual branches. This allows us to accurately isolate the contribution of the selected action $a$ from the state value:

$$A(s, a) = V(s_a) - V(s), \tag{7}$$

where $V(s_a)$ is the value of the child node resulting from action $a$. We then adopt a PPO-style (Schulman et al., 2017) clipped objective function to update the policy network using this high-quality advantage estimate.

To avoid the heuristic tuning of the fixed entropy coefficient, we incorporate an **adaptive entropy regularization** mechanism (Haarnoja et al., 2018). We treat the temperature parameter $\alpha$ as a learnable variable and formulate the optimization as a constrained problem, aiming to maximize the expected return while satisfying a minimum entropy constraint. The joint optimization objective is defined as:

$$\begin{aligned} \mathcal{L}_{\text{TAPO}}(\theta) = \mathbb{E}_{(s,a)\sim\mathcal{D}} \Big[ &\min \Big( r_t(a) A(s, a), \\ &\text{clip}\big(r_t(a), 1-\epsilon, 1+\epsilon\big) A(s, a) \Big) \Big] + \alpha H(\pi_\theta(\cdot|s)). \end{aligned} \tag{8}$$

Simultaneously, we update $\alpha$ to ensure the policy entropy covers the desired *target entropy* $\mathcal{H}_{\text{target}}$. The loss function for $\alpha$ is given by:

$$\mathcal{L}(\alpha) = \mathbb{E}_{s\sim\mathcal{D}}[-\alpha(\log \pi_\theta(a|s) + \mathcal{H}_{\text{target}})], \tag{9}$$

where $\mathcal{H}_{\text{target}}$ is a hyperparameter representing the lower bound of the expected entropy. By minimizing $\mathcal{L}(\alpha)$, the temperature $\alpha$ automatically increases when the policy becomes too deterministic and decreases when exploration is sufficient, thereby maintaining a dynamic balance between exploration and exploitation throughout the training process.

The core contribution of this approach lies in the resolution of the credit assignment challenge. By leveraging the counterfactual baselines established via tree search, our method enables the policy to accurately estimate the marginal contribution of selecting a specific model at critical DNs. Consequently, the router learns to discern specifically how the choice of the next token affects the long-term computational cost and correctness of the final answer.

This tree-advantage signal differs from both supervised routing and standard rollout-based RL. Supervised token routers typically learn local labels extracted from a reference trajectory, which does not directly evaluate how a local routing action changes the future autoregressive prefix. A GRPO-style rollout update instead assigns actions a trajectory-level return, which remains shared across all decisions on the sampled path. TAPO first backs up leaf returns into tree node values and then updates each expanded DN using the sibling-branch value gap $V(s_a) - V(s)$ under the same prefix. This converts delayed trajectory rewards into localized credit-assignment signals at the exact states where SLM and LLM choices diverge.

### 4.5. Node Identification

After completing the optimization of the DN router $\pi_{div}$ via TAPO, we proceed to the training of the node identifier $\pi_{node}$. We approach this as a supervised binary classification task, where the goal is to distinguish between ANs and DNs based on the current state.

To construct the training dataset, we freeze the parameters of the trained $\pi_{div}$ and perform inference on the dataset following the collaborative path extension described in Section 4.4.1. For each sample, we generate an inference trajectory and record the node type at each step as the ground truth label $y$. Specifically, states where the SLM and LLM produce identical tokens are labeled as ANs ($y = 0$), while states exhibiting disagreement are labeled as DNs ($y = 1$).

We collect these state-label pairs into a dataset $\mathcal{D}_{cls}$ and optimize $\pi_{node}$ by minimizing the Binary Cross-Entropy Loss:

$$\begin{aligned} \mathcal{L}_{cls}(\theta) = - \mathbb{E}_{(s,y)\sim\mathcal{D}_{cls}} \Big[ &y \log \sigma(\pi_\theta(s)) \\ &+ (1-y) \log(1 - \sigma(\pi_\theta(s))) \Big], \end{aligned} \tag{10}$$

where $f_\theta(s)$ denotes the output logits of the classification head, and $\sigma$ is the Sigmoid activation function. Through this process, $\pi_{node}$ effectively learns to acquire the consistency verification capability, enabling the system to identify reliable SLM-dominant states early in the inference process without needing to invoke the LLM for comparison during the final deployment. The pseudocode of AlphaRouter is provided in Appendix A.2.

## 5. Experiments

### 5.1. Setup

**Baselines.** We compare AlphaRouter against three categories of models: (1) **Static Baselines**: The Qwen2.5-Instruct family (Qwen et al., 2025), including the SLM (0.5B), LLM (7B), and intermediate sizes (1.5B, 3B) to benchmark scaling efficiency; (2) Query-Level Routers: Representative methods from RouteLLM (QR-MF, QR-BERT, and QR-LLM) (Ong et al., 2025); and (3) Token-

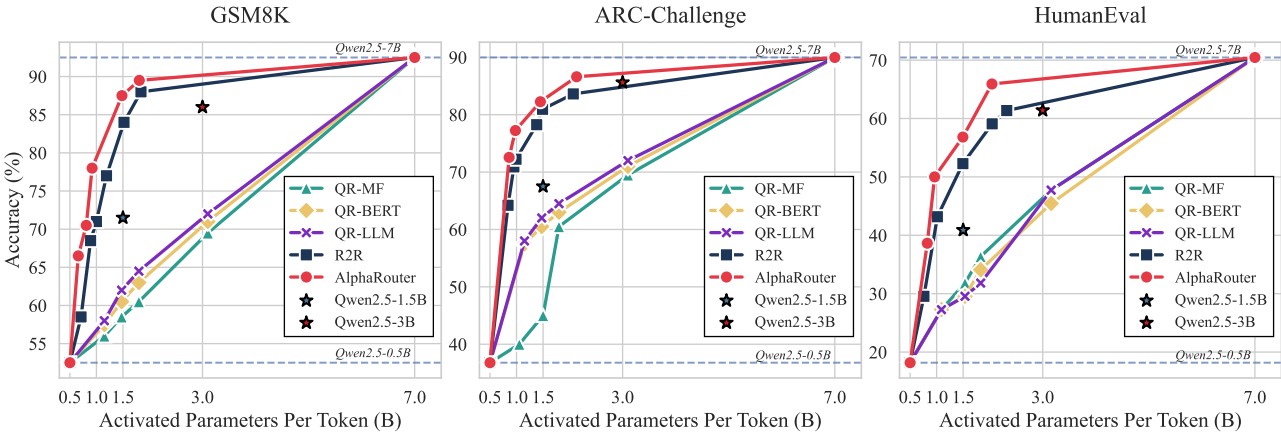

*Figure 3.* **Accuracy versus efficiency curves across three benchmarks.** The x-axis represents the average activated parameters per token (in Billions), and the y-axis represents the accuracy (%). The plots compare AlphaRouter (red curve) against static Qwen2.5 checkpoints (black stars) and various routing baselines (QR-MF, QR-BERT, QR-LLM, R2R).

*Table 1.* **Performance comparison on GSM8K, ARC-Challenge, and HumanEval.** The table reports the Accuracy (Acc., %), LLM Call Rate (CR., %), and Average Activated Parameters (Param., in Billions) for static models, query-level routers, and token-level routers. "Type" indicates the method category, and baselines include different sizes of Qwen2.5 and the RouteLLM/R2R frameworks.

| Type | Method | GSM8K | | | ARC-Challenge | | | HumanEval | | | Average | | |
|---|---|---|---|---|---|---|---|---|---|---|---|---|---|
| | | Acc. | CR. | Param. | Acc. | CR. | Param. | Acc. | CR. | Param. | Acc. | CR. | Param. |
| SLM | Qwen2.5-0.5B | 52.5% | 0.0% | 0.5 | 36.8% | 0.0% | 0.5 | 18.2% | 0.0% | 0.5 | 36.8% | 0.0% | 0.5 |
| LLM | Qwen2.5-7B | 92.5% | 100.0% | 7.0 | 90.0% | 100.0% | 7.0 | 70.5% | 100.0% | 7.0 | 84.3% | 100% | 7.0 |
| **Ours** | **AlphaRouter** | **90.5%** | **22.5%** | **2.0** | **86.6%** | **25.1%** | **2.1** | **65.9%** | **23.7%** | **2.0** | **81.0%** | **23.8%** | **2.0** |
| | QR-MF | 56.0% | 10.0% | 1.2 | 40.0% | 8.5% | 1.1 | 27.3% | 9.1% | 1.1 | 41.1% | 9.2% | 1.1 |
| | QR-BERT | 57.5% | 10.0% | 1.2 | 39.0% | 9.1% | 1.1 | 27.3% | 9.1% | 1.1 | 41.3% | 9.4% | 1.1 |
| 1B | QR-LLM | 58.0% | 10.0% | 1.2 | 38.0% | 7.0% | 1.0 | 27.3% | 9.1% | 1.1 | 41.1% | 8.7% | 1.1 |
| | R2R | 71.0% | 7.7% | 1.0 | 72.6% | 7.6% | 1.0 | 43.2% | 7.9% | 1.0 | 62.3% | 7.8% | 1.0 |
| | **AlphaRouter** | **78.0%** | **6.5%** | **0.9** | **77.2%** | **7.4%** | **1.0** | **50.0%** | **7.1%** | **1.0** | **68.4%** | **7.0%** | **1.0** |
| | Qwen2.5-1.5B | 71.5% | 15.4% | 1.5 | 67.5% | 15.4% | 1.5 | 40.9% | 15.4% | 1.5 | 61.3% | 15.4% | 1.5 |
| | QR-MF | 58.5% | 15.0% | 1.5 | 45.0% | 15.4% | 1.5 | 31.8% | 15.9% | 1.5 | 45.1% | 15.4% | 1.5 |
| 1.5B | QR-BERT | 60.5% | 15.0% | 1.5 | 42.5% | 15.4% | 1.5 | 29.6% | 15.9% | 1.5 | 44.2% | 15.4% | 1.5 |
| | QR-LLM | 62.0% | 15.0% | 1.5 | 42.0% | 14.0% | 1.4 | 29.6% | 15.9% | 1.5 | 44.5% | 15.0% | 1.5 |
| | R2R | 84.0% | 15.7% | 1.5 | 81.0% | 15.2% | 1.5 | 47.7% | 15.3% | 1.5 | 70.7% | 15.4% | 1.5 |
| | **AlphaRouter** | **87.5%** | **15.1%** | **1.5** | **82.3%** | **14.6%** | **1.4** | **54.6%** | **15.3%** | **1.5** | **74.8%** | **15.0%** | **1.5** |
| 3B | Qwen2.5-3B | 86.0% | 38.5% | 3.0 | 85.6% | 38.5% | 3.0 | 61.4% | 38.5% | 3.0 | 77.7% | 38.5% | 3.0 |

Level Baselines: The state-of-the-art method R2R (Fu et al., 2025).

**Datasets & Implementation.** We conduct zero-shot evaluations on GSM8K (Math) (Cobbe et al., 2021), **ARC-Challenge** (Reasoning) (Clark et al., 2018), and **HumanEval** (Coding) (Chen, 2021). AlphaRouter is trained to dynamically route tokens between Qwen2.5-0.5B and Qwen2.5-7B, with a tunable threshold to control the computational budget.

**Metrics.** We evaluate performance using **Accuracy** (pass@1 for code) and computational cost using **Average Activated Parameters (AAP)** per token, a hardware-agnostic metric derived from the weighted usage of the SLM and LLM. Since AAP does not capture systems effects such as model switching and LLM prefilling, we also report end-to-end latency, throughput, computation, memory traffic, and peak memory in Appendix C.2.

Further details concerning the experimental setup can be found in Appendix B.

### 5.2. Performance

**Superior Performance across Benchmarks.** Table 1 presents a comprehensive evaluation on GSM8K, ARC-Challenge, and HumanEval, where AlphaRouter demonstrates a consistent and dominant advantage over both static baselines and state-of-the-art routing methods. In the constrained low-resource regime (approx. 1.0B parameters), our method establishes a significant lead. For instance, on the reasoning-intensive GSM8K benchmark, AlphaRouter

achieves a remarkable accuracy of **78.0%** while activating only **0.9B** parameters per token on average. This performance surpasses the strongest token-level baseline, R2R (71.0%), by a substantial margin of **7.0%**, while notably maintaining a lower LLM Call Rate (CR) of 6.45% compared to R2R's 7.7%. The advantage extends to general knowledge tasks; on ARC-Challenge, AlphaRouter attains **82.3%** accuracy with an average cost of 1.4B parameters. This decisively outperforms the monolithic Qwen2.5-1.5B model, which achieves only 67.5% accuracy despite consuming higher computational resources. Even on the code generation task HumanEval, which typically challenges SLM, AlphaRouter maintains its superiority, achieving **65.9%** accuracy with 2.0B parameters, effectively bridging the gap between efficient inference and high-performance coding capabilities. Notably, while activating only 2.0B parameters, our method achieves higher accuracy than the 3B model and closely approaches the performance of the 7B model (with a gap of only 3.3%).

**Advancing the Pareto Frontier.** Figure 3 visually quantifies the trade-off between inference accuracy and computational cost. The x-axis denotes the average activated parameters per token, while the y-axis represents the achieved task accuracy. As illustrated by the distinct red curve, AlphaRouter effectively **pushes the evaluated Pareto frontier** towards the optimal top-left region, improving the accuracy–cost trade-off over the baselines across the tested computational budgets. Specifically, it outperforms Query-Level routers (QR-MF, QR-BERT, QR-LLM) by overcoming their inherent granularity limitations, where routing entire prompts to smaller models leads to catastrophic failures on hard instances. Furthermore, it surpasses the previous state-of-the-art token-level method, R2R, demonstrating that our search-based policy explores the solution space more effectively than direct supervised cloning.

Crucially, the trajectory of AlphaRouter consistently lies significantly above the discrete checkpoints of static models (marked as black stars). For instance, on ARC-Challenge, our method achieves performance comparable to the 7B model while utilizing less than 30% of its active parameters. This geometric dominance validates a core finding: dynamic token-level collaboration yields a fundamentally higher return on computation than simply scaling up model size. Instead of relying on uniform parameter scaling, AlphaRouter identifies and leverages regions of maximal marginal utility, selectively allocating heavy computation only to those high-leverage tokens that are decisive for the final outcome. This allows the system to bypass the constraints of traditional scaling laws, achieving disproportionate performance gains relative to the added computational cost.

**Generalization across LLM Sizes.** We further investigate the zero-shot transferability of AlphaRouter by applying the policy—trained solely on the *0.5B+7B* pair—directly to other unseen SLM-LLM combinations without any fine-tuning. Since the router relies on the context output by the SLM as input, the SLM cannot be replaced for a fixed router. Therefore, to verify the generalization capability of the router, we instead replace the LLM with unseen 3B and 14B models for experimentation. Detailed metrics are provided in Table 7 (see Appendix C.3). The routing policy transfers effectively across the tested backend LLM sizes while keeping the SLM fixed. When transferred to the **0.5B+3B** pair, the system achieves an accuracy of **90.0%**. This result is counter-intuitive and significant: the collaborative pair outperforms the standalone LLM Qwen2.5-3B model (86.0%) by **4.0%**. This supports our observation that collaborative trajectories can sometimes improve over a standalone greedy LLM trajectory, rather than merely imitating it. Furthermore, when scaling up to the **0.5B+14B** pair, AlphaRouter demonstrates strong efficiency, achieving **87.5%** accuracy with only **2.11B** average activated parameters. This corresponds to roughly **15%** of the computational cost of the standalone Qwen2.5-14B, while retaining the vast majority of its performance capabilities. These findings suggest that AlphaRouter can reuse a fixed SLM-aware router across different backend LLM scales, while cross-SLM transfer remains an important limitation discussed in Appendix D.

### 5.3. System-Level Efficiency

AAP is useful for hardware-agnostic comparison, but it does not fully capture real deployment cost because invoking the LLM may require prefilling tokens that were generated by the SLM. We therefore profile end-to-end generation on a single NVIDIA A800 80GB GPU over 100 randomly sampled GSM8K requests; the measurements include routing overhead, model switching, and LLM prefilling. Table 2 summarizes representative operating points, with the full statistics and memory breakdown in Appendix C.2.

*Table 2.* System-level efficiency on GSM8K. Latency and throughput are measured end-to-end and include switching and prefilling overhead.

| Model | Acc. | Latency (s) | TPS | TFLOPs | Mem. (TB) |
|---|---|---|---|---|---|
| Qwen-7B | 91% | 3.8420 | 88.00 | 588.19 | 682.10 |
| Spec. Decoding | 91% | 2.5886 | 132.82 | 605.64 | 708.46 |
| **AlphaRouter** | **86%** | **2.5582** | **134.46** | **237.25** | **281.97** |
| **AlphaRouter** | **91%** | 2.9459 | 115.06 | 259.92 | 305.06 |

At the matched 91% accuracy point, AlphaRouter is faster than standalone Qwen-7B (2.9459s vs. 3.8420s) while reducing both computation and memory traffic. In particular, it lowers total computation from 588.19 to 259.92 TFLOPs, a 55.8% reduction, and lowers total memory access from

682.10 to 305.06 TB, a 55.3% reduction. These gains are important because autoregressive decoding is often memory-bandwidth bound: AlphaRouter improves latency not only by activating fewer parameters, but also by substantially reducing memory movement per request.

Speculative decoding remains a strong latency baseline at this high-accuracy point, but it preserves the target model behavior and therefore does not expose the same budget-adjustable accuracy–efficiency frontier. It also increases total computation and memory access relative to standalone Qwen-7B (605.64 vs. 588.19 TFLOPs; 708.46 vs. 682.10 TB) due to draft-and-verify overhead. By contrast, AlphaRouter at the same 91% accuracy uses only 42.9% of SD's computation and 43.1% of its memory access, while the 86% operating point further reduces the cost to 237.25 TFLOPs and 281.97 TB. AlphaRouter therefore provides controllable operating points: one can trade moderate accuracy for lower latency and substantially lower memory traffic, or move toward the high-accuracy regime without changing the backbone models.

### 5.4. Ablation Study

**Effectiveness of TAPO-optimized Routing Policy.** We compare AlphaRouter with **AlphaRouter-Diff**, which uses the same disagreement labels but routes to the LLM whenever the identifier predicts an SLM–LLM mismatch. As shown in Table 8 (Appendix C.4), AlphaRouter consistently outperforms this heuristic, especially under the strict 1B budget (**78.00%** vs. 64.50%). This 13.5% gap shows that *not all predicted disagreements warrant correction*: TAPO learns to spend calls on consequential divergences, reaching higher accuracy with a lower LLM call rate (6.45% vs. 8.80%).

**Contribution of Tree-structured Credit Assignment.** We further compare AlphaRouter with **AlphaRouter-GRPO**, a variant that replaces TAPO's sibling-branch advantage with a standard rollout objective. Appendix C.4 shows that AlphaRouter performs better across all tested budgets, with the largest gain in the low-call-rate regime. Thus, CITS is more than a data collection mechanism: its tree structure supplies localized counterfactual baselines for sparse routing decisions.

**Impact of CITS Hyperparameters.** We also study the CITS path budget $M$ and expansion rounds $N$ in Table 9 (Appendix C.5). Reducing $M$ from 32 to 16 lowers accuracy from **90.50%** to **86.00%**, indicating that diverse candidate trajectories are important for low-variance advantage estimates. Increasing $N$ from 2 to 3 also improves performance (e.g., 89.50% to 90.50% at $M = 32$), suggesting that deeper exploration better captures long-term routing effects. These offline search choices improve the learned policy without

increasing inference latency or parameter cost ($\approx$1.96B).

## 6. Conclusion

In this work, we proposed **AlphaRouter**, a framework that transforms token-level routing from static imitation to a dynamic *search-and-optimize* paradigm. By formalizing collaborative inference as an MDP and introducing Collaborative Inference Tree Search (CITS) alongside Tree-Advantage Policy Optimization (TAPO), we effectively resolved the credit assignment challenge in sparse-reward inference tasks. Extensive experiments show that AlphaRouter improves the evaluated Pareto frontier, achieving performance comparable to 7B models with only $\approx$2.0B active parameters. Additional system-level measurements further show that these gains are not limited to AAP: AlphaRouter can reduce end-to-end latency, computation, and memory traffic under realistic switching and prefilling overhead. Furthermore, transfer across tested backend LLM sizes suggests that a fixed SLM-aware router can support flexible deployment across different large-model budgets. Ablation studies further demonstrate the necessity of tree search over divergent nodes and the role of TAPO in assigning credit to consequential routing decisions.

## Acknowledgements

The above work was supported in part by the Joint Funds of the National Natural Science Foundation of China under Grant U25A20436, the National Natural Science Foundation of China (NSFC) (62372047), Guangxi Key Research & Development Program (FN2504240036, 2025FN96441087), Guangdong S&T Programme (No. 2025B0101120006), the Natural Science Foundation of Guangdong Province (2024A1515011323), the Supplemental Funds for Major Scientific Research Projects of Beijing Normal University, Zhuhai (ZHPT2023002), the Fundamental Research Funds for the Central Universities, Guangdong Province Educational Science Planning Research Project under 2025JKZG069, Higher Education Research Topics of Guangdong Association of Higher Education in the 14th Five-Year Plan under 24GYB207.

## Impact Statement

The development of AlphaRouter as a framework for token-level collaborative inference has the potential to significantly impact the efficiency of large language model deployment. There are many potential societal consequences of our work, none of which we feel must be specifically highlighted here.

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

# A. Algorithm pseudocode

## A.1. Collaborative Inference Tree Search

---

**Algorithm 1** Budget-Constrained Collaborative Inference Tree Search (CITS)

---

**Require:** Dataset $\mathcal{D}$, SLM $\pi_s$, LLM $\pi_l$, DN Router $\pi_{div}$
**Require:** Total path budget $M$, Max expansion rounds $N$
1: Initialize tree $\mathcal{T}$ with input prompts from $\mathcal{D}$
2: Initialize global candidate set $\mathcal{C} \leftarrow \emptyset$
3: $\mathcal{P}_{curr} \leftarrow$ Run **CollaborativePathExtension**(Roots of $\mathcal{T}$)
4: **for** round $t = 1$ to $N$ **do**
5:     Calculate remaining budget: $M_{remain} \leftarrow M - |\mathcal{P}_{curr}|$
6:     **if** $M_{remain} \leq 0$ **then**
7:         **break**
8:     **end if**
9:     Calculate expansion budget: $K_t \leftarrow \lceil M_{remain}/(N - t + 1) \rceil$
10:     ▷ *Phase 1: Candidate Sampling*
11:     Select expansion nodes: $\mathcal{V}_{exp} \leftarrow \text{Sample}(\mathcal{C}, \min(K_t, |\mathcal{C}|))$
12:     **for** each node $v \in \mathcal{V}_{exp}$ **do**
13:         Identify existing action $a$ taken at $v$
14:         Force counterfactual action: $a' \leftarrow \neg a$
15:         Create new branch starting with action $a'$
16:         ▷ *Phase 2: Extend new branch to completion*
17:         $\mathcal{P}_{new} \leftarrow$ **CollaborativePathExtension**(New Branch)
18:         $\mathcal{P}_{curr} \leftarrow \mathcal{P}_{curr} \cup \mathcal{P}_{new}$
19:         Remove $v$ from $\mathcal{C}$
20:     **end for**
21: **end for**
22: **return** Collaborative Inference Tree $\mathcal{T}$

23: **function CollaborativePathExtension**(Active Nodes)
24:     **for** each step in generation **do**
25:         Get next tokens: $y_s \sim \pi_s$, $y_l \sim \pi_l$
26:         **if** $y_s = y_l$ **then**
27:           Mark as **Agree Node (AN)**
28:           Append $y_s$ to prefix
29:         **else**
30:           Mark as **Divergent Node (DN)**
31:           Add node to $\mathcal{C}$
32:           Sample action $a \sim \pi_{div}(\cdot|s)$
33:           Append token corresponding to $a$
34:         **end if**
35:     **end for**
36:     **return** Completed trajectories
37: **end function**

---

## A.2. AlphaRouter Training Framework

---

**Algorithm 2** AlphaRouter Training Framework

---

**Require:** SLM $\pi_s$, LLM $\pi_l$, Reward function $\mathcal{R}$
**Require:** Initial DN Router $\pi_{div}^{\theta}$, Node Identifier $\pi_{node}^{\phi}$
**Require:** Hyperparameters: $\lambda$, $\epsilon$, $\mathcal{H}_{\text{target}}$
1: **Phase 1: Train DN Router via TAPO**
2: **while** not converged **do**
3:     ▷ *Step 1: Data Collection via CITS*

4:      $\mathcal{T} \leftarrow \text{CITS}(\mathcal{D}, \pi_s, \pi_l, \pi_{div}^\theta)$
5:      ▷ *Step 2: Tree-backup Value Estimation*
6:      **for** each node $s$ in $\mathcal{T}$ (Post-order Traversal) **do**
7:        **if** $s$ is leaf **then**
8:          $V(s) \leftarrow r_T$
9:        **else if** $|\text{children}(s)| = 1$ **then**
10:          ▷ *Inherit child value*
11:          $V(s) \leftarrow V(s')$
12:        **else**
13:          ▷ *DN with counterfactual branches*
14:          $V(s) \leftarrow \pi_{div}^\theta(a_s|s)V(s_s) + \pi_{div}^\theta(a_l|s)V(s_l)$
15:        **end if**
16:      **end for**
17:      ▷ *Step 3: Policy Optimization*
18:      **for** each expanded DN $(s, a)$ in $\mathcal{T}$ **do**
19:        Calculate Advantage: $A(s, a) \leftarrow V(s_a) - V(s)$
20:        Update $\theta$ by maximizing $\mathcal{L}_{\text{TAPO}}(\theta)$ (Eq. 8)
21:        Update entropy temperature $\alpha$ minimizing $\mathcal{L}(\alpha)$ (Eq. 9)
22:      **end for**
23: **end while**
24: **Phase 2: Train Node Identifier**
25: Freeze $\pi_{div}^\theta$
26: Initialize dataset $\mathcal{D}_{cls} \leftarrow \emptyset$
27: **for** each input $x \in \mathcal{D}$ **do**
28:      Run inference using $\pi_s, \pi_l, \pi_{div}^\theta$
29:      **for** each step $t$ **do**
30:        **if** $y_s = y_l$ **then**
31:          Label $y \leftarrow 0$ (AN)
32:        **else**
33:          Label $y \leftarrow 1$ (DN)
34:        **end if**
35:        Add $(s_t, y)$ to $\mathcal{D}_{cls}$
36:      **end for**
37: **end for**
38: Update $\phi$ by minimizing $\mathcal{L}_{cls}(\phi)$ (Eq. 10)
39: **return** Optimized $\pi_{div}^\theta, \pi_{node}^\phi$

# B. Experimental Setup Details

### B.1. Baselines

We compare AlphaRouter against a comprehensive set of baselines spanning three distinct categories to validate its effectiveness across different granularities and scaling paradigms.

**Static Baselines.** We utilize the Qwen2.5-Instruct family (Qwen et al., 2025) to establish the performance bounds and verify the efficiency of dynamic collaboration against static scaling.

- **SLM (Qwen2.5-0.5B-Instruct):** The backbone model for our router, serving as the lower bound for performance and computational cost.

- **LLM (Qwen2.5-7B-Instruct):** The collaborative expert model, serving as the upper bound for performance.

- **Intermediate Models (1.5B, 3B):** We include Qwen2.5-1.5B and Qwen2.5-3B to represent the "static scaling law." These checkpoints allow us to verify whether AlphaRouter achieves a better trade-off than simply training a medium-sized monolithic model.

**Query-Level Routers.** We compare against representative methods from the RouteLLM framework (Ong et al., 2025), which route the entire prompt to either the SLM or LLM. This comparison highlights the advantage of token-level granularity.

- **QR-MF (Matrix Factorization):** A cost-effective router that learns a projection from the query embedding to model performance scores.

- **QR-BERT:** A BERT-based classifier trained to predict the optimal model for a given prompt based on human preference data.

- **QR-LLM:** Uses a Llama3-8B-based LLM (Grattafiori et al., 2024) prompted to select the backend model.

**Token-Level Baselines.** We compare against **R2R (Routing to Reasoning)** (Fu et al., 2025), the current state-of-the-art token-level routing method. R2R employs a supervised learning approach to clone the behavior of a teacher LLM. Comparing against R2R allows us to isolate the specific contribution of our search-based RL (TAPO) approach versus standard supervised cloning.

**Baseline Choice.** We do not include Co-LLM (Shen et al., 2024) and CITER (Zheng et al., 2025) as primary baselines because a key practical advantage of AlphaRouter is that it trains only lightweight routing modules on top of fixed backbone models. In contrast, the released pipelines of Co-LLM and CITER involve additional model adaptation beyond the router itself. Co-LLM includes base-model adaptation during deferral training, while CITER first adapts the model on reasoning data and then uses the adapted model to construct token-level labels or preferences for router training. We therefore use R2R as the most directly comparable token-level routing baseline under a fixed-backbone setting.

**Evaluation Protocol.** CITS is used only during offline training. At test time, AlphaRouter produces a single routed trajectory for each input and does not run tree search, select among multiple paths, or evaluate multiple candidate outputs. Accordingly, HumanEval is evaluated as standard pass@1: unit tests are run only on the single final output generated by the deployed router.

## B.2. Datasets and Tasks

We evaluate our method on three benchmarks covering diverse reasoning capabilities: **GSM8K** (Mathematical QA), **ARC-Challenge** (General Reasoning), and **HumanEval** (Code Generation). Table 3 summarizes the statistical specifications for each dataset, including the total sample count and the standard splits for training and testing.

*Table 3.* Statistics of the datasets used in our experiments. Note that for HumanEval, we utilize the standard release which consists solely of evaluation samples. For ARC-Challenge, we evaluate in a zero-shot setting using the test set.

| Dataset | Task Type | Total Samples | Train Set | Test Set |
| --- | --- | --- | --- | --- |
| **GSM8K** (Cobbe et al., 2021) | Math Reasoning | 1,200 | 1,000 | 200 |
| **ARC-Challenge** (Clark et al., 2018) | General QA | 1,299 | 1,000 | 299 |
| **HumanEval** (Chen, 2021) | Code Generation | 164 | 120 | 44 |

- **GSM8K:** A dataset of high-quality grade school math word problems. We use the standard training set (approx. 7.5k examples) to construct the CIT search space and train the router, and evaluate on the test set. We use Chain-of-Thought (CoT) prompting for all models.

- **ARC-Challenge:** A multiple-choice question set focusing on grade-school science questions that require reasoning. Although the dataset provides a training split, we evaluate exclusively in a zero-shot setting on the test set to verify the router's generalization to knowledge-intensive tasks.

- **HumanEval:** A set of Python coding problems. We assess the models using the *pass@1* metric. This benchmark is purely evaluation-based, consisting of 164 hand-written programming problems. This task is particularly challenging for routers as a single incorrect token can break the entire code execution.

## B.3. Implementation Details

**Model Architecture.** For the AlphaRouter implementation, we utilize Qwen2.5-0.5B-Instruct as the base SLM and Qwen2.5-7B-Instruct as the collaborative LLM. Unlike standard probes that rely solely on hidden states, our routing modules ($\pi_{node}$ and $\pi_{div}$) employ a **multi-source feature fusion** strategy. The input representation is constructed by concatenating the SLM's last hidden states, the embedding of the current token, and the projected features of the top-$k$ ($k = 100$ default) logits, capturing both semantic context and model confidence. Architecturally, both policies are implemented using a Residual MLP Backbone (He et al., 2016) composed of Pre-LayerNorm followed by an MLP with an expansion factor of 4 and GELU activation. To balance capacity and efficiency, we adopt an asymmetric design: the **Node Identifier** ($\pi_{node}$)—responsible for the nuanced task of consistency verification—is deeper, consisting of **6 residual blocks** with a hidden dimension of 1024. In contrast, the **Divergent Router** ($\pi_{div}$) is designed for speed, utilizing a shallower architecture with **2 residual blocks** of the same width. Despite the increased depth of the identifier, the total parameter count remains minimal compared to the backbone LLMs, ensuring that the routing overhead remains negligible ($< 0.1\%$ of the SLM inference latency).

**Training Hyperparameters.** We train the router using the AdamW optimizer. For **CITS**, we set the total path budget $M = 32$ and the maximum expansion rounds $K = 3$ (as analyzed in the Ablation Study). For **TAPO**, we use a learning rate of $1 \times 10^{-5}$, a batch size of 64, a target KL 0.02, and a PPO clip range $\epsilon = 0.2$. The target entropy $\mathcal{H}_{\text{target}}$ is set to 0.15 to encourage deterministic decisions after convergence. The inference latency penalty $\lambda$ in the reward function is setting at 0.01.

**Hardware.** Main benchmark experiments are conducted on $2 \times$ NVIDIA A100 (80GB) GPUs. The system-level profiling and training-cost breakdown in Appendix C.2 are measured on a single NVIDIA A800 (80GB) GPU. The collaborative inference framework is implemented using PyTorch and SGLang (Zheng et al., 2024) for accelerated token generation.

## B.4. Evaluation Metrics

To ensure a hardware-agnostic and fair comparison, we focus on two primary metrics:

**Accuracy.** For GSM8K and ARC-Challenge, we report the standard accuracy (exact match of the final answer). For HumanEval, we report *pass@1*, which measures the percentage of problems where the generated code passes all unit tests.

**Average Activated Parameters (AAP).** Since actual latency depends heavily on hardware optimization and system load, we use AAP as a proxy for computational cost. It represents the expected number of parameters activated to generate a single token. Formally, let $P_{SLM}$ and $P_{LLM}$ be the parameter counts of the SLM and LLM, respectively, and $CR$ be the LLM Call Rate (percentage of tokens routed to the LLM). The AAP is calculated as:

$$\text{AAP} = P_{SLM} \times (1 - \text{CR}) + P_{LLM} \times \text{CR}. \tag{11}$$

Note that the SLM is always active (to generate the prefix context and router features). This metric penalizes unnecessary LLM calls and provides a linear scale for efficiency comparison.

## B.5. Prompts

In this section, we illustrate the prompt we used for each dataset.

### B.5.1. GSM8K

> **Prompt Template for GSM8K**
>
> ```
> Please solve the following math problem with detailed steps and put your
> final answer within \boxed{}.
> ```
> **Problem:** `{Input Question}`

B.5.2. ARC-CHALLENGE

---

**Prompt Template for ARC-Challenge**

```
Please think step by step and finally answer with the option letter in the
format 'Answer: (X)'.

Question: {Input Question}

Answer Choices:
A. {Answer choice}
B. {Answer choice}
C. {Answer choice}
D. {Answer choice}
```

---

B.5.3. HUMANEVAL

---

**Prompt Template for HumanEval**

```
You are a Python coding expert.  Let's think step by step first, and then
output the code.
You should Provide the complete, self-contained code for the following
function request.  Include imports if necessary.  Do not output any test
case.

Code: {Input Code}
```

---

## C. Experiment Results

### C.1. Training Cost and CITS Parallelism

CITS is an offline training mechanism rather than a test-time search procedure. In our implementation, tree construction is *frontier-parallel but round-synchronous*: active paths in the same expansion round are decoded in batched form, after which the algorithm synchronizes to update the global DN candidate set and select counterfactual branches for the next round. Thus, CITS is not a fully serial tree expansion, but it also incurs synchronization overhead between expansion rounds.

*Table 4.* Training cost breakdown for AlphaRouter on a single NVIDIA A800 80GB GPU.

| Phase | Latency (h) | #GPU | GPU Hour |
|---|---|---|---|
| CITS Rollout | 5.26 | 1 | 5.26 |
| DN Router Training | 0.29 | 1 | 0.29 |
| Node Type Sampling | 0.25 | 1 | 0.25 |
| DN Identifier Training | 0.04 | 1 | 0.04 |
| Total | 5.84 | 1 | 5.84 |

Most of the training overhead comes from the CITS rollout stage. This cost is paid once during training and then distilled into the deployed node identifier and DN router, which perform single-path autoregressive routing at inference time.

### C.2. Inference Efficiency and System-Level Cost Analysis

We measure real end-to-end inference cost on a single NVIDIA A800 80GB GPU by randomly sampling 100 GSM8K problems. These wall-clock measurements include routing overhead, model switching, and LLM prefilling cost. In addition

to latency and throughput, we report total computation and memory access because autoregressive decoding is often memory-bandwidth bound.

*Table 5.* Inference efficiency and performance comparison.

| Model | CR. | Param. (B) | Avg Tokens | Avg Latency (s) | Avg TPS | Acc. | TFLOPs | Total Memory Access (TB) | Mem/Token (GB) |
|---|---|---|---|---|---|---|---|---|---|
| Qwen-0.5B | – | 0.49 | 330.29 | 0.6789 | 485.35 | 45% | 32.37 | 43.57 | 1.319 |
| Qwen-1.5B | – | 1.54 | 341.83 | 2.0207 | 169.38 | 73% | 120.15 | 139.62 | 4.085 |
| Qwen-7B | – | 7.62 | 338.01 | 3.8420 | 88.00 | 91% | 588.19 | 682.10 | 20.180 |
| SD | – | 8.10 | 326.95 | 2.5886 | 132.82 | 91% | 605.64 | 708.46 | 21.669 |
| **AlphaRouter** | 8.74% | 1.07 | 321.98 | 2.0821 | 165.14 | 78% | 212.45 | 253.21 | 7.864 |
| **AlphaRouter** | 14.95% | 1.47 | 316.53 | 2.5582 | 134.46 | 86% | 237.25 | 281.97 | 8.908 |
| **AlphaRouter** | 18.70% | 1.72 | 328.97 | 2.9459 | 115.06 | 91% | 259.92 | 305.06 | 9.375 |

At the matched 91% accuracy point, AlphaRouter reduces latency from 3.8420s to 2.9459s relative to Qwen-7B, while lowering computation from 588.19 TFLOPs to 259.92 TFLOPs and memory access from 682.10 TB to 305.06 TB. Speculative decoding (SD) is faster at this same accuracy point, but incurs higher computation and memory traffic than Qwen-7B because it runs an additional draft-and-verification process. AlphaRouter is complementary: it exposes multiple accuracy–efficiency operating points through its routing threshold, including lower-cost regimes that are not naturally available to SD.

*Table 6.* Attributable memory footprint during offline training and deployment.

| Component | Train Mem. (GiB) | Train % | Test Mem. (GiB) | Test % |
|---|---|---|---|---|
| 7B model weights | 14.18 | 91.0% | 14.18 | 92.3% |
| 0.5B model weights | 0.92 | 5.9% | 0.92 | 6.0% |
| Router weights | 0.22 | 1.4% | 0.22 | 1.4% |
| Peak 7B KV cache | 0.24 | 1.5% | 0.04 | 0.3% |
| Peak 0.5B KV cache | 0.05 | 0.3% | 0.01 | 0.1% |
| Total attributable memory | 15.61 | 100% | 15.37 | 100% |

At deployment, AlphaRouter keeps both the 0.5B and 7B models resident and maintains KV caches for both. The additional footprint is dominated by the small model and router weights: relative to the standalone 7B footprint of approximately 14.22 GiB from 7B weights and KV cache, AlphaRouter uses about 15.37 GiB in this profiling setting, an increase of 1.15 GiB. This overhead is real but moderate compared with the 7B model itself, and is similar in nature to other two-model collaborative inference methods.

## C.3. Generalization across LLM Sizes

*Table 7.* **Generalization capability of AlphaRouter across different model scales.** We evaluate the transferability of the routing policy trained on the *0.5B+7B* pair to other LLM sizes (3B and 14B) without re-training. **CR.** denotes LLM Call Rate, and **Param.** represents Average Activated Parameters in Billions.

| Type | Models | Description | Acc. (%) | CR. (%) | Param. (B) |
|---|---|---|---|---|---|
| SLM | Qwen2.5-0.5B | Reference Baseline | 52.50 | - | 0.50 |
| LLM | Qwen2.5-3B | Reference Baseline | 86.00 | - | 3.00 |
| | Qwen2.5-7B | Reference Baseline | 92.50 | - | 7.00 |
| | Qwen2.5-14B | Reference Baseline | 93.50 | - | 14.00 |
| **AlphaRouter** | 0.5B + 3B | Generalized from 0.5B+7B | **90.00** | 48.32 | 1.71 |
| | 0.5B + 7B | **Trained on 0.5B+7B** | **90.50** | **22.52** | **1.96** |
| | 0.5B + 14B | Generalized from 0.5B+7B | 87.50 | 11.96 | 2.11 |

## C.4. Routing Policy Ablations

We evaluate the contribution of TAPO from two complementary perspectives. First, **AlphaRouter-Diff** uses the same DN identifier training setup as AlphaRouter, but removes the TAPO-optimized routing policy: during inference, it routes to the LLM whenever the identifier predicts a divergent node. This baseline therefore does not compare the SLM and LLM outputs online; it tests whether a learned disagreement detector alone is sufficient. Second, **AlphaRouter-GRPO** replaces TAPO's sibling-branch advantage with a standard rollout-based RL objective, testing whether the gain comes merely from using RL rather than from tree-structured credit assignment.

*Table 8.* **Routing policy ablations on GSM8K. Acc.** denotes Accuracy, **CR.** denotes LLM Call Rate, and **Param.** represents Average Activated Parameters (in Billions).

| Comparison | Type | Model | Acc. (%) | CR. (%) | Param. (B) |
|---|---|---|---|---|---|
| Heuristic DN routing | 1B | AlphaRouter-Diff | 64.50 | 8.80 | 1.07 |
| | | **AlphaRouter** | **78.00** | **6.45** | **0.92** |
| | 1.5B | AlphaRouter-Diff | 74.50 | 15.95 | 1.54 |
| | | **AlphaRouter** | **87.50** | **15.10** | **1.48** |
| | 2B | AlphaRouter-Diff | 87.00 | 24.55 | 2.10 |
| | | **AlphaRouter** | **90.50** | **22.52** | **1.96** |
| Rollout-based RL | 1B | AlphaRouter-GRPO | 72.50 | 8.58 | 1.06 |
| | | **AlphaRouter** | **78.00** | **6.45** | **0.92** |
| | 1.5B | AlphaRouter-GRPO | 84.00 | 15.96 | 1.54 |
| | | **AlphaRouter** | **87.50** | **15.10** | **1.48** |
| | 1.8B | AlphaRouter-GRPO | 88.50 | 19.84 | 1.79 |
| | | **AlphaRouter** | **90.00** | **18.87** | **1.73** |

The AlphaRouter-Diff comparison shows that not every predicted model disagreement should trigger the LLM: naive DN routing wastes budget on benign differences and performs substantially worse. The AlphaRouter-GRPO comparison shows that replacing TAPO with a trajectory-level rollout objective also degrades performance, especially in the low-call-rate regime. Together, these results indicate that AlphaRouter's gain comes from using tree-backed sibling comparisons to assign credit at consequential DNs, rather than simply detecting disagreement or applying generic RL.

## C.5. Impact of CITS Hyperparameters

*Table 9.* **Ablation study on CITS hyperparameters.** We investigate the impact of the maximum number of preserved paths ($M$) and the number of expansion rounds ($K$) on the routing performance. The evaluation is conducted on the GSM8K dataset. **Param.** denotes Average Activated Parameters (Billions).

| Total Path ($M$) | Exp. Rounds ($K$) | Acc. (%) | CR. (%) | Param. (B) |
|---|---|---|---|---|
| 32 | 3 | **90.50** | 22.52 | 1.96 |
| | 2 | 89.50 | 22.49 | 1.96 |
| 16 | 3 | 86.00 | **22.13** | **1.94** |
| | 2 | 84.50 | 23.41 | 2.02 |

# D. Limitations

AlphaRouter is studied in the standard token-level collaborative inference setting where both the SLM and LLM are available at deployment time. This enables flexible routing but also requires keeping both models resident, so the peak memory footprint is slightly larger than that of the standalone LLM. In addition, the current router uses SLM hidden states and logits as input, making it SLM-aware rather than directly transferable across different SLM checkpoints without retraining or adaptation.

Our experiments mainly evaluate greedy decoding and task-specific router training. Extending AlphaRouter to stochastic

decoding settings, broader cross-task training mixtures, and cross-SLM adaptation remains important future work.

# E. Examples

In this section, we present a representative GSM8K example comparing AlphaRouter with standalone Qwen2.5-0.5B and Qwen2.5-7B. For AlphaRouter, red highlights indicate spans generated by the LLM.

---

**GSM8K Example**

**Question.** A certain company is in the business of selling fresh fruit. One crate of such fruit consists of 5 bananas, 12 apples, and 7 oranges. The price for such a crate depends on the price of its individual fruits. One apple costs $0.5 and one banana costs twice as much. Oranges are the most expensive and cost three times as much as a banana per piece. What would be the price for such a crate of fruit?
**Gold answer.** $32.

---

**Qwen2.5-0.5B-Instruct**          Incorrect

To determine the price of a crate of fruit, we need to calculate the cost of each type of fruit and then sum these costs. Let's break it down step by step.

1. **Calculate the cost of one banana:**

   - One banana costs $0.5.
   - Therefore, the cost of one banana is $0.5.

2. **Calculate the cost of one apple:**

   - One apple costs twice as much as one banana.
   - Therefore, the cost of one apple is $2 \times 0.5 = 1$ dollar.

3. **Calculate the cost of one orange:**

   - One orange costs three times as much as one banana.
   - Therefore, the cost of one orange is $3 \times 0.5 = 1.5$ dollars.

4. **Calculate the total cost of one crate of fruit:**

   - One crate consists of 5 bananas, 12 apples, and 7 oranges.
   - The cost of the bananas is $5 \times 0.5 = 2.5$ dollars.
   - The cost of the apples is $12 \times 1 = 12$ dollars.
   - The cost of the oranges is $7 \times 1.5 = 10.5$ dollars.
   - Therefore, the total cost of one crate is $2.5 + 12 + 10.5 = 25$ dollars.

So, the price for such a crate of fruit is $\boxed{25}$ dollars.

---

**Qwen2.5-7B-Instruct**          Correct

To solve this problem, we need to calculate the total cost of the crate of fruit by determining the cost of each type of fruit and then summing these costs.
First, let's define the cost of each type of fruit:

- The cost of one apple is given as $0.5.

- The cost of one banana is twice the cost of one apple, so it is $2 \times 0.5 = 1$ dollar.

- The cost of one orange is three times the cost of one banana, so it is $3 \times 1 = 3$ dollars.

Next, we calculate the total cost for each type of fruit in the crate:

---

- There are 5 bananas in the crate, so the total cost for the bananas is $5 \times 1 = 5$ dollars.

- There are 12 apples in the crate, so the total cost for the apples is $12 \times 0.5 = 6$ dollars.

- There are 7 oranges in the crate, so the total cost for the oranges is $7 \times 3 = 21$ dollars.

Finally, we sum the costs of all the fruits to find the total cost of the crate:

$$5 \text{ (bananas)} + 6 \text{ (apples)} + 21 \text{ (oranges)} = 32 \text{ dollars}$$

Thus, the price for such a crate of fruit is $\boxed{32}$.

---

**AlphaRouter**        **Correct**

To determine the price of a crate of fruit, we need to calculate the cost of each type of fruit and then sum these costs. Let's break it down step by step.

1. **Calculate the cost of one banana:**

    - One banana costs twice as much as one apple.
    - One apple costs $0.5.
    - Therefore, the cost of one banana is $2 \times 0.5 = 1$ dollar.

2. **Calculate the cost of one orange:**

    - One orange costs three times as much as one banana.
    - One banana costs $1 .
    - Therefore, the cost of one orange is $3 \times 1 = 3$ dollars.

3. **Calculate the total cost of one crate of fruit:**

    - One crate consists of 5 bananas, 12 apples, and 7 oranges.
    - The cost of 5 bananas is $5 \times 1 = 5$ dollars.
    - The cost of 12 apples is $12 \times 0.5 = 6$ dollars.
    - The cost of 7 oranges is $7 \times 3 = 21$ dollars.
    - Therefore, the total cost of one crate is $5 + 6 + 21 = 32$ dollars.

So, the price for such a crate of fruit is $\boxed{32}$ dollars.

