# OpenReview forum: "AlphaRouter: Token-level Routing Between SLM and LLM with Reinforcement Learning and Tree Search"
_ICML.cc/2026/Conference — ICML 2026 regular_

### Official Review · Reviewer_Qtkw · 2026-03-07

**Soundness:** 2
**Presentation:** 3
**Significance:** 3
**Originality:** 3
**Overall Recommendation:** 3
**Confidence:** 4

**Summary:**

This paper studies token-level routing between a small language model (SLM) and a large language model (LLM) to improve the cost–performance trade-off during inference. Instead of learning routing policies by imitating LLM trajectories, the authors propose AlphaRouter, which learns routing decisions via reinforcement learning with Collaborative Inference Tree Search (CITS). The search explores alternative routing trajectories when SLM and LLM predictions diverge, producing a collaborative inference tree. A PPO-style algorithm (Tree‑Advantage Policy Optimization, TAPO) uses value estimates from the tree to update the routing policy. At inference time, a lightweight router predicts whether the models would diverge and decides which model should generate each token. Experiments on GSM8K, ARC, and HumanEval show improvements over several routing baselines under similar compute budgets measured by average activated parameters.

**Compliance With Llm Reviewing Policy:**

Affirmed.

**Final Justification:**

The rebuttal improves the paper’s empirical completeness, particularly by adding end-to-end latency/throughput results and a more informative ablation against a standard RL-style variant.

However, my main concerns remain unresolved. The paper still does not provide empirical evidence for cross-task or cross-domain generalization without retraining, and the router’s reliance on SLM-specific representations limits transferability across SLMs. While the rebuttal clarifies these points, they remain important limitations that affect the generality and practical applicability of the method.

Overall, I appreciate the additional experiments and clarifications, but since the core concerns are not addressed empirically, my overall assessment and score remain unchanged.

**Key Questions For Authors:**

1. If the router is trained on one task (e.g., GSM8K), how does it perform on a different domain such as coding or general QA without retraining? Have the authors tested cross-task generalization?
2. What is the total training cost of AlphaRouter (e.g., GPU hours, number of LLM calls, or tokens processed during CITS)? This would help assess the practicality of the approach.
3. Can the authors report actual inference latency or throughput instead of only average activated parameters? Model switching and routing overhead could affect real-world speed.
4. What happens if the router is trained using standard RL rollouts without CITS? This would clarify the contribution of the collaborative inference tree.
5. Since the router depends on SLM representations, have the authors evaluated whether the router can transfer to a slightly different SLM (e.g., same architecture but different size or checkpoint)?
6. How does the method compare with widely used acceleration strategies such as speculative decoding or draft‑verify frameworks in terms of both accuracy and latency?

**Limitations:**

Yes.

**Strengths And Weaknesses:**

## Strengths

- The paper proposes learning routing policies through search and RL rather than imitation of LLM outputs, which is conceptually appealing.
- The idea of branching only when SLM and LLM outputs diverge is a neat way to explore routing trajectories while controlling search cost.
- Experiments show consistent improvements over several heuristic and RL routing methods at similar compute budgets.

## Weaknesses

- The router effectively learns when the SLM is likely to fail on the training distribution. The paper does not evaluate whether the learned routing policy generalizes across domains or tasks.
- The router relies on SLM hidden states and logits, meaning the SLM cannot be replaced without retraining the router.
- The method requires tree search with repeated SLM/LLM inference during training, but the paper does not report training compute or time.
- The paper uses average activated parameters as a proxy for efficiency but does not report actual latency or throughput.
- It is unclear how much improvement comes from CITS versus the RL training framework itself.

---

> ### Author Rebuttal · Authors · 2026-03-31
>
> We thank the reviewer for the thoughtful and well-structured feedback.
>
> **W1 & Q1:** We agree that broader transfer evaluation would strengthen the paper. At present, our results support **zero-shot generalization to unseen LLM scales**, but we do not yet claim broad cross-task generalization without retraining. We will make this scope explicit and avoid overstating generality. We will also add cross-task transfer as an important future direction.
>
> **W2 & Q5:** We agree. Our current router relies on SLM hidden states and logits, so replacing the SLM may indeed require retraining or adaptation. We will state this limitation explicitly in the revised paper and discuss transfer across nearby SLM variants as an important future direction.
>
> **W3 & Q2:** We agree that this should be reported more clearly. We have now added a detailed training-cost breakdown on a single A800 80GB GPU. To avoid repetition, please see our response to **Reviewer zsVZ (W1)** for the full table.
>
> **W4 & Q3:** We agree. We have therefore added **end-to-end latency and throughput measurements**, together with computation and memory-access statistics, on a single A800 80GB GPU over 100 randomly sampled GSM8K requests. These results show that AlphaRouter can match Qwen-7B’s 91% accuracy with lower latency and significantly lower memory traffic, while also exposing a **more flexible accuracy–efficiency trade-off** than speculative decoding. Please see our detailed response to **Reviewer 9tUx (W1 & Q1)** for the full table and analysis.
>
> **W5 & Q4:** We agree that isolating the contribution of **CITS** is important. To better address this point, we added an ablation against a **GRPO-style** alternative that removes our tree-advantage learning design and replaces it with a more standard RL-style objective. The results are shown below:
>
> | Type | Model            | Acc. (%)  | CR. (%)   | Param. (B) |
> | ---- | ---------------- | --------- | --------- | ---------- |
> | 1B   | AlphaRouter-GRPO | 72.50     | 8.58      | 1.06       |
> |      | **AlphaRouter**  | **78.00** | **6.45**  | **0.92**   |
> | 1.5B | AlphaRouter-GRPO | 84.00     | 15.96     | 1.54       |
> |      | **AlphaRouter**  | **87.50** | **15.10** | **1.48**   |
> | 1.8B | AlphaRouter-GRPO | 88.50     | 19.84     | 1.79       |
> |      | **AlphaRouter**  | **90.00** | **18.87** | **1.73**   |
>
> Across all tested scales, AlphaRouter consistently outperforms the GRPO-style variant, with the advantage being **most pronounced in the low-call-rate regime**, where routing quality matters most. At 1B, AlphaRouter improves accuracy from **72.5% to 78.0%** while also reducing CR (**8.58% → 6.45%**) and activated parameters (**1.06B → 0.92B**). At 1.5B, the gain remains clear (**84.0% → 87.5%**) with slightly lower CR and parameters. At 1.8B, the gap becomes smaller, which is expected as both methods move into a higher-call-rate regime where the routing problem is easier.
>
> We believe this pattern shows that the gain does **not** come merely from “using RL,” but from the **tree-structured training signal enabled by CITS**, which is especially valuable under tighter LLM budgets. CITS exposes counterfactual collaborative trajectories and provides richer credit assignment than a standard rollout-based objective. If we replace this tree-advantage mechanism with a conventional RL objective, performance degrades most clearly in the regime where routing quality is most critical. This suggests that CITS is a key component rather than an implementation detail.
>
> **Q6:** This is a useful suggestion. Conceptually, our method targets a different problem: speculative decoding is primarily a **systems acceleration** technique that preserves the target model’s output, whereas AlphaRouter learns a **task-driven token-level decision policy** over a collaborative search space, which can in some settings find better trajectories than the greedy LLM path. That said, we agree that the comparison is relevant from an efficiency perspective, so we have included a speculative-decoding baseline in the runtime table; please see **Reviewer 9tUx (W1 & Q1)**. We will clarify in the paper that these approaches are complementary rather than strictly interchangeable.
>
> We thank the reviewer again for carefully considering our rebuttal, and we hope the additional analyses help clarify the scope, efficiency, and contribution of our method.

---

> > ### Author Rebuttal · Reviewer_Qtkw · 2026-04-03
> >
> > Thank you for the detailed rebuttal. The additional results substantially improve the paper’s completeness, especially regarding efficiency evaluation (e.g., end-to-end latency/throughput) and the added ablation comparing against a more standard RL-style variant.
> >
> > However, I do not think my core concerns are fully resolved. In particular, the rebuttal does not provide empirical evidence for cross-task or cross-domain generalization without retraining, and it also confirms that the router remains tied to SLM-specific hidden states/logits, which limits transferability across SLMs. In my view, these limitations still significantly affect the generality and practical applicability of the method.
> >
> > Overall, I appreciate the authors’ clarifications and additional experiments, and I agree that the rebuttal strengthens the paper. However, since the main concerns about cross-task generalization and cross-SLM portability remain unaddressed empirically, I will keep my original score.

---

> > > ### Author Response · Authors · 2026-04-07
> > >
> > > Thank you for the thoughtful follow-up. We agree that cross-task generalization and cross-SLM portability are important directions for extending this line of work.
> > >
> > > For cross-SLM transfer, our router is intentionally **SLM-aware** rather than SLM-agnostic: it relies on the SLM’s hidden states and logits because routing decisions depend on the SLM’s token-level uncertainty and failure patterns. As a result, replacing the SLM changes the representation space on which the router operates, so direct transfer across SLMs is generally not expected. In our view, this is not a limitation unique to AlphaRouter, but a broader structural trade-off shared by token-level routing methods (e.g., CITER, Co-LLM, and R2R) that rely on model-specific internal representations.
> > >
> > > For cross-task generalization, we agree that the current rebuttal does not yet provide a direct empirical study across multiple domains without retraining, and we will make this scope explicit in the final version. At the same time, we would like to clarify that AlphaRouter can be trained relatively quickly once a task category is specified, making such adaptation practical. More broadly, we believe cross-task generalization is primarily determined by training diversity rather than a restriction of the routing framework itself: if trained on a mixture of diverse datasets spanning multiple task types, AlphaRouter is expected to learn routing patterns that generalize across tasks. We will clarify this point and, where possible, include additional multi-task results in the final version.
> > >
> > > We hope this helps distinguish between the current experimental scope and the broader extensibility of the method. If you find these updates helpful, we would greatly appreciate it if you would be open to reconsidering your score.

---

### Official Review · Reviewer_9tUx · 2026-03-08

**Soundness:** 2
**Presentation:** 1
**Significance:** 2
**Originality:** 2
**Overall Recommendation:** 4
**Confidence:** 2

**Summary:**

The paper proposes a tree-based framework for SLM-LLM token routing. Given the same prefix context, the authors define Agree Nodes (AN), where the SLM and LLM agree, and Diverge Nodes (DN), where they disagree. The framework trains a Node Identifier (binary classification) and a Node Router (sequential decision problem) to allocate compute dynamically, utilizing Tree-Advantage Policy Optimization (TAPO) for routing decisions.

**Compliance With Llm Reviewing Policy:**

Affirmed.

**Final Justification:**

The authors have adequately addressed several of my initial concerns, and I have raised my score from 2 to 4 accordingly.

However, my reservation regarding the additional memory overhead caused by the SLM's KV cache during the training rollout phase remains unresolved. The peak VRAM figures reported in the rebuttal (0.24 GiB for the 7B and 0.05 GiB for the 0.5B model) appear artificially low because they are derived from a limited subset of 100 GSM8k samples. This evaluation inherently involves short context/short reasoning horizons and small batch sizes.

In realistic training scenarios requiring longer generation horizons and larger batch sizes, the memory costs associated with maintaining active paths and counterfactual branches during the CITS rollout phase will scale substantially. Due to this lingering uncertainty regarding the framework's scalability and some presentation issues, I have lowered my confidence score.

**Key Questions For Authors:**

Q1: Please provide the actual wall-clock inference runtime (e.g., tokens per second), number of generated tokens (by SLM and LLM separately) and peak memory usage, comparing the proposed framework against standalone SLM and LLM generation across multiple values of $M$ and $N$.

Q2: Regarding the pass@1 code benchmarks: Does your evaluation run unit tests against all $M$ generated paths and count a success if any pass (effectively pass@$M$)? If you do select a single trajectory for a true pass@1 evaluation, what exact mechanism collapses the tree to choose that final sequence before testing?

Q3: How does Algorithm 1 handle hardware parallelism? Can the $M$ paths be generated simultaneously, or does managing the DN pool force sequential execution and inefficient GPU batching?

Q4: How does this framework scale to reasoning-heavy tasks requiring long Chain-of-Thought (CoT)? Have you considered evaluating against a best-of-$M$ or pass@$M$ baseline for standalone models to ensure a fair comparison of the expanded solution space?

I am open to update my reviews or add more questions throughout the rebuttal period.

**Limitations:**

limitations are not discussed.

**Strengths And Weaknesses:**

Strength:
- The paper explores a relevant and interesting research topic.

Weakness:
- The reported metrics (Call Rate, AAP) are deceptive proxies. Having a low AAP and LLM call rate does not accurately reflect actual runtime in real-world deployments.
- The primary goal of SLM-LLM routing is to accelerate inference. However, the framework's inference process (e.g., Algorithm 1) appears to operate sequentially. If Algorithm 1 cannot be parallelized, it will introduce a significant computational bottleneck, undermining the main advantage of SLM-LLM collaborative inference. It is unclear why users would accept a 2-5% drop in accuracy for potentially longer inference times.
- In the introduction, the paper establishes a compelling premise: "we observe that for the vast majority of cases where the LLM performs poorly, there exist better trajectories in the collaborative inference space." Yet, the evidence that collaborative inference surpasses a standalone LLM only exists for the 0.5B and 3B model pair, while performance actually decreases by 2-5% for the 0.5B/7B and 0.5B/14B pairs. This limited evidence does not sufficiently support the bold claim (lines 421-423, first column) that the premise is confirmed.
- Lines 312-314 claim to use the pass@1 metric for code benchmarks. I assume (though I cannot find explicit clarification in the paper) that the baselines and standalone LLM generate and evaluate the correctness of a single trajectory. If your proposed method generates M inference trajectories across N rounds, this comparison is fundamentally unfair. The authors should clarify this or switch to a fairer accuracy metric.

---

> ### Author Rebuttal · Authors · 2026-03-31
>
> We thank the reviewer for the detailed and constructive feedback. We agree that the original draft did not make the system-level story sufficiently explicit, and we appreciate the opportunity to clarify it.
>
> **W1 & Q1:** We agree that Call Rate and AAP are only proxy metrics and do not fully capture deployment cost. We have therefore added **actual end-to-end inference measurements**, including wall-clock latency, throughput, computation, and memory traffic, all measured on a single A800 80GB GPU over 100 randomly sampled GSM8K requests. These results already include practical switching overhead, including LLM prefilling cost. The key finding is that, at the matched 91% accuracy point, AlphaRouter is still faster than Qwen-7B while also requiring substantially less computation and memory access. The results also clarify the trade-off against **speculative decoding (SD)**: SD achieves very strong latency at the 91% point, but with **higher computation and memory traffic** than standalone Qwen-7B, whereas AlphaRouter exposes a more **flexible accuracy–efficiency frontier**, allowing operation at much lower-cost regimes with only moderate accuracy sacrifice. Since autoregressive decoding is largely memory-bound, this reduction in memory traffic is especially important for practical throughput. To avoid repetition, please see our detailed response to **Reviewer vNzc (W1 & Q2)** for the full runtime table and analysis.
>
> **W2 & Q3:** We agree that the current draft did not clearly distinguish **training-time search** from **deployment-time inference**. In our framework, CITS is used **only during training** as an offline exploration mechanism to construct counterfactual routing trajectories and optimize the DN router via TAPO. At deployment time, AlphaRouter does **not** perform tree expansion or maintain multiple active paths. Instead, after the DN router is trained, we further train a lightweight node identifier so that the deployed system can predict likely agreement/divergence states and perform **single-path autoregressive routing**. Therefore, the deployed inference pipeline is **not** a sequential online tree-search process. We will revise Algorithm 1 and the surrounding text to make this distinction explicit.
>
> **W3:** We agree that our wording should be more careful. Our intended claim is **not** that collaborative routing always surpasses the standalone LLM for every model pair. Rather, our empirical observation is that collaborative trajectories can, in some settings, discover solutions that the greedy LLM trajectory misses. We will tone down the wording accordingly.
>
> **W4 & Q2:** Yes — HumanEval is evaluated under **true pass@1**. Since CITS is used only during training, AlphaRouter produces exactly **one final routed trajectory per problem** at test time. We do **not** execute unit tests over multiple searched paths and count success if any one of them passes. Unit tests are run only on the single final output, consistent with the standard pass@1 protocol.
>
> **Q4:** AlphaRouter is **not** intended as a best-of-k or pass@k style test-time method. Any additional search cost is incurred during **training**, where CITS is used offline to improve the routing policy. At test time, the deployed policy follows a **single routed path**. In this sense, AlphaRouter shifts extra computation to **training time**, whereas best-of-k / pass@k methods increase **inference-time** compute through multi-sample decoding. We will revise the paper to make this distinction more explicit.
>
> We are grateful to the reviewer for taking the time to read our rebuttal, and we hope the added clarifications make the deployment setting and evaluation protocol clearer.

---

> > ### Author Rebuttal · Reviewer_9tUx · 2026-04-03
> >
> > Additional questions:
> >
> > Q1. The rebuttal clarifies that Algorithm 1 (CITS) is an offline training mechanism, but the core question regarding its hardware parallelization remains unanswered. Managing the global candidate set of Divergent Nodes appears to force sequential execution or inefficient GPU batching. Can these expansion paths be generated simultaneously? Please provide the actual training overhead (wall-clock time and peak memory) required to generate these counterfactual trees.
> >
> > Q2. The provided metrics demonstrate a reduction in average memory traffic (bandwidth) per token. However, this metric masks the impact on peak memory capacity. Because AlphaRouter must simultaneously store the model weights and continuously track the expanding KV caches for both the 0.5B and 7B models, the total memory footprint is strictly higher than running the standalone 7B model. How does this dual-cache requirement impact peak VRAM usage during deployment? Will AlphaRouter hit Out of Memory (OOM) errors faster than the standalone LLM on benchmarks requiring long Chain-of-Thought (CoT) reasoning?

---

> > > ### Author Response · Authors · 2026-04-07
> > >
> > > Thank you for the helpful follow-up. We will clarify the hardware behavior of CITS and peak memory overhead more explicitly.
> > >
> > > First, **CITS is not fully sequential, but it is also not globally asynchronous**. In our implementation, parallelism happens at the **frontier level**: all currently active paths in the same expansion round are decoded together in batched form. After that round finishes, the algorithm synchronizes, builds the global set of divergent nodes, selects which ones to expand, and launches those new branches in the next round. Therefore, the correct characterization is **frontier-parallel but round-synchronous**, rather than fully serial tree expansion.
> > >
> > > Second, **KV cache is reused along each active path**, since decoding is incremental rather than restarted from scratch at every token. However, newly created counterfactual branches are **not explicitly materialized by cloning the parent branch’s cache** in our implementation; branch-level reuse is handled implicitly by the backend cache manager. Thus, the method benefits from KV reuse within active paths, while branch expansion still incurs synchronization overhead between rounds.
> > >
> > > Regarding the actual overhead of generating these counterfactual trees, we measured it in our offline training pipeline on a single NVIDIA A800 80GB GPU. The wall-clock breakdown has been reported in our response to **Reviewer zsVZ**: the full AlphaRouter training cost is **5.84 GPU-hours** in total, including **5.26h** for CITS rollout, **0.29h** for DN router training, **0.25h** for node type sampling, and **0.04h** for DN identifier training.
> > >
> > > We also logged the attributable memory footprint during both offline training and deployment:
> > >
> > > | Component                     | Train-time Memory (GiB) | Train-time % | Test-time Memory (GiB) | Test-time % |
> > > | ----------------------------- | ----------------------- | ------------ | ---------------------- | ----------- |
> > > | 7B model weights              | 14.18                   | 91.0%        | 14.18                  | 92.3%       |
> > > | 0.5B model weights            | 0.92                    | 5.9%         | 0.92                   | 6.0%        |
> > > | router weights                | 0.22                    | 1.4%         | 0.22                   | 1.4%        |
> > > | peak 7B KV cache              | 0.24                    | 1.5%         | 0.04                   | 0.3%        |
> > > | peak 0.5B KV cache            | 0.05                    | 0.3%         | 0.01                   | 0.1%        |
> > > | **Total attributable memory** | **15.61**               | **100%**     | **15.37**              | **100%**    |
> > >
> > > This also helps clarify the reviewer’s concern about peak VRAM. At deployment, AlphaRouter keeps both the 0.5B and 7B models resident and maintains KV caches for both. However, in our profiling, the additional footprint from the 0.5B side is small compared with the 7B model itself: at test time, the 7B model uses **14.18 GiB** for weights and **0.04 GiB** for KV cache, while the 0.5B model uses only **0.92 GiB** for weights and **0.01 GiB** for KV cache. Therefore, the deployment footprint remains close to the standalone 7B case. Relative to a standalone 7B baseline (about **14.22 GiB** from 7B weights + 7B KV), AlphaRouter uses about **15.37 GiB** of attributable test-time memory, i.e., an increase of only **1.15 GiB**. For long CoT decoding, this overhead is mainly due to the additional small model and router, rather than test-time tree materialization or unusually large KV-cache growth. This is also broadly consistent with other **two-model collaborative inference methods**, such as speculative decoding, which likewise keep both a small and a large model resident during decoding.
> > >
> > > If you find these updates helpful, we would greatly appreciate it if you would be open to reconsidering your score.

---

### Official Review · Reviewer_zsVZ · 2026-03-12

**Soundness:** 2
**Presentation:** 3
**Significance:** 3
**Originality:** 2
**Overall Recommendation:** 4
**Confidence:** 4

**Summary:**

This paper proposes AlphaRouter, a token-level routing framework that decides whether to use an SLM or LLM at each decoding step. The key insight is that the collaborative SLM+LLM space can find correct answers even when the LLM alone fails. The AlphaRouter framework introduces a Collaborative Inference Tree Search with a Tree-based policy optimization method. Experiments show AlphaRouter achieves on-par performance with 7B models with only 2B average activated parameters.

**Compliance With Llm Reviewing Policy:**

Affirmed.

**Final Justification:**

My concerns have been addressed, and my overall evaluation of the submission has changed.

**Key Questions For Authors:**

N/A

**Limitations:**

There is no discussion on limitations. For suggestions please see weaknesses.

**Strengths And Weaknesses:**

Strengths
- The core observation is valuable: collaborative inference can outperform the LLM alone by exploring counterfactual paths.
- Strong experimental results; AlphaRouter dominates the Pareto frontier across three benchmarks and generalizes zero-shot to unseen LLM sizes .
- The paper is generally well-written and clear to follow.

Weaknesses
- Training cost is not shown in this paper. Since the algorithm requires running two models at the same time while using tree search during training, the training cost is likely significantly more expensive compared to the baseline.
- While the performance is good in terms of Pareto frontier, it is still not a breakthrough with a typical ~3.5x parameter efficiency. This is especially important since the method still requires two models to be run simultaneously locally.
- The paper is restricted to greedy sampling. Many applications use temperature sampling.
- R2R is supervised learning; AlphaRouter is RL with expensive tree search. Controlling for training compute would be fairer.
- There is a lack of technical analysis.

---

> ### Author Rebuttal · Authors · 2026-03-31
>
> We thank the reviewer for the thoughtful feedback and for recognizing both the value of the core idea and the strength of the empirical results.
>
> **W1 & W4:** We agree that the original submission did not report training cost clearly enough. We have now added a detailed training-cost breakdown for AlphaRouter:
>
> | Model       | Phase                  | Latency (hours) | #GPU  | GPU Hour |
> | ----------- | ---------------------- | --------------- | ----- | -------- |
> | AlphaRouter | CITS Rollout           | 5.26            | 1     | 5.26     |
> |             | DN Router Training     | 0.29            | 1     | 0.29     |
> |             | Node Type Sampling     | 0.25            | 1     | 0.25     |
> |             | DN Identifier Training | 0.04            | 1     | 0.04     |
> |             | **Total**              | **5.84**        | **1** | **5.84** |
>
> All training was conducted on a single NVIDIA A800 80GB GPU. The total cost is **5.84 GPU-hours**, with most of the overhead coming from the CITS rollout stage.
>
> Regarding baselines, we agree this point should be discussed more explicitly. While R2R is a supervised token-level router and does not use our collaborative tree-search training procedure, its training/data-generation pipeline is also non-trivial. The R2R paper reports generating 7.6M routing labels from 2,094 queries using R1-1.5B and R1-32B, requiring about **2.3 days on 8 A800 GPUs** for data generation alone; the router is then trained on top of this labeled dataset.
>
> That said, we agree that a direct training-cost comparison under the same hardware and setting would be more informative. Due to time constraints during the rebuttal period, we have not yet completed a unified re-measurement of R2R training time under our hardware/configuration. We will add this controlled comparison in the final version so that the trade-off between AlphaRouter and supervised routing baselines such as R2R is presented more clearly and fairly.
>
> **W2:** We agree that activated-parameter savings alone are not sufficient, and we also agree that our method, like other token-level routing approaches, requires access to both an SLM and an LLM at deployment time. We will make this point clearer in the revision.
>
> That said, we would like to emphasize two practical advantages of AlphaRouter. First, within this two-model setting, AlphaRouter achieves a stronger **cost–accuracy trade-off** than simply using a distilled small model alone. Under matched or comparable activated-parameter budgets, AlphaRouter attains higher accuracy while also delivering faster end-to-end inference and substantially lower memory access than similarly sized standalone small models (see our response to **Reviewer 9tUx, W1 & Q1**). For example, compared with Qwen-1.5B, AlphaRouter achieves substantially higher accuracy at similar latency, while maintaining much lower memory traffic than Qwen-7B.
>
> Second, the two-model setup is not merely a deployment burden; it also provides useful **controllability**. By adjusting the routing threshold, AlphaRouter can move along a **Pareto frontier** and flexibly trade accuracy for cost at inference time, which is valuable across application scenarios with different latency or resource budgets. In our experiments, AlphaRouter can operate at multiple points—from lower-cost / lower-latency regimes to a high-accuracy regime that matches Qwen-7B—without changing the backbone models. This flexibility is difficult to obtain from a single distilled model, whose operating point is largely fixed once trained.
>
> In short, we agree that deploying two models is a real system consideration, but this is inherent to the token-level routing setting rather than unique to AlphaRouter. The key question is whether the added flexibility and improved Pareto trade-off justify that design choice, and our results suggest that they do.
>
> **W3:** We agree that the current study focuses on greedy decoding, and we will explicitly note this in the limitations section. At the same time, we would like to clarify that this setting is not specific to our method: our experiments follow the same greedy-decoding setup commonly used in prior token-level routing / collaboration work, including R2R, CITER, and Co-LLM. Therefore, the comparison is made under a shared and controlled evaluation protocol, rather than under a setting chosen specifically for AlphaRouter.
>
> **W5:** We agree that more analysis is helpful. We therefore added an ablation against a GRPO-style alternative to isolate the contribution of our tree-advantage learning design. Across all tested scales, the full AlphaRouter consistently outperforms the GRPO-style variant. To avoid repetition, please see our response to **Reviewer Qtkw (W5 & Q4)** for the full ablation table.
>
> We sincerely appreciate the reviewer taking the time to consider our rebuttal, and we hope the added clarifications help address the concerns raised above.

---

> > ### Author Rebuttal · Reviewer_zsVZ · 2026-04-01
> >
> > In W2, the authors agree that using two models is not ideal, though they did not provide any solution or potential remedies.
> >
> > In W5, I was referring to theoretical analysis on how and why this proposed scheme can work, not just case studies or ablations.

---

> > > ### Author Response · Authors · 2026-04-02
> > >
> > > Thank you for the helpful follow-up, and we are grateful for the opportunity to clarify these details more explicitly.
> > >
> > > **W2:** We would like to clarify that our rebuttal did not intend to suggest that “using two models is not ideal” in itself. Since our problem setting is **token-level SLM–LLM collaborative inference**, the availability of both models is part of the setup rather than a flaw specific to our method. This is also the standard setting in prior token-level collaboration/routing works such as **Co-LLM, CITER, and R2R**. Our point was only that this creates a real system-level consideration; accordingly, the relevant question is whether the router can substantially reduce expensive LLM usage and deliver a better accuracy–efficiency trade-off within this collaborative setting.
> > >
> > > **W5:** We agree that a more technical explanation is needed, especially in comparison with both the **supervised routing line** (e.g., R2R, CITER, and Co-LLM) and a direct **GRPO-style RL** update. In our view, the key distinction is the following: supervised routing learns from **token-level labels derived from a reference route**; GRPO optimizes **trajectory-level preference using sampled returns**; by contrast, TAPO first converts trajectory returns into **tree-backed node values**, and then performs **branch-level credit assignment at divergent nodes**.
> > >
> > > Compared with supervised routing, the key limitation is that existing supervised approaches typically treat token-level routing as learning from **local decision labels** extracted from a reference trajectory. However, under autoregressive generation, the effect of a routing action is inherently **non-local**: choosing the SLM or LLM at step $t$ changes the next token, which changes the future prefix, and therefore changes the distribution of subsequent tokens and potentially the final answer. As a result, the quality of a routing decision at step $t$ cannot, in general, be determined solely by whether it matches a reference model’s local choice; it should instead be evaluated by the **long-horizon return of the branch induced by that choice**. In this sense, the limitation of the supervised routing line is not merely that it uses labels, but that it does not fully account for how a local routing decision propagates through the autoregressive process and affects downstream generation quality.
> > >
> > > For a sampled trajectory $\tau$, a direct GRPO-style update uses a trajectory-level signal of the form
> > >  $A^{\mathrm{GRPO}}(s,a)\approx R(\tau_a)-b(s),$
> > >  where $R(\tau_a)$ denotes the final return of the sampled trajectory containing action $a$, and $b(s)$ is a baseline. Thus, the update at node $s$ is still driven by the quality of the **entire sampled trajectories**, rather than by an explicit comparison between the two sibling branches at that specific divergent node.
> > >
> > > By contrast, TAPO does not use the sampled return at a DN directly. Instead, it first assigns each leaf node its trajectory return,
> > >  $V(\mathrm{leaf}) = R(\tau),$
> > >  and then performs **tree backup**. For a node with a single child, $V(s)=V(s')$; for an expanded divergent node,
> > >  $V(s)=(1-p)V(s_s)+pV(s_l),$
> > >  where $p=\pi_\theta(l\mid s)$, and $s_s,s_l$ denote the child states induced by the SLM and LLM branches, respectively. TAPO then defines
> > >  $A^{\mathrm{TAPO}}(s,a)=V(s_a)-V(s).$
> > >  Therefore,
> > >  $A^{\mathrm{TAPO}}(s,l)=(1-p)\bigl(V(s_l)-V(s_s)\bigr),\qquad A^{\mathrm{TAPO}}(s,s)=-p\bigl(V(s_l)-V(s_s)\bigr).$
> > >
> > > This makes the distinction explicit. **Supervised routing** in prior work relies on token-level labels from a reference route; **GRPO** updates from a sampled trajectory return $R(\tau)$; **TAPO** updates from a sibling-branch value gap derived from tree-backed node values $V$. Equivalently,
> > >  $\frac{\partial V(s)}{\partial p}=V(s_l)-V(s_s),$
> > >  which shows that TAPO reallocates probability according to which branch has the higher long-horizon value under the same prefix. In this sense, TAPO uses the tree structure to transform delayed trajectory returns into **localized credit-assignment signals** at divergence points.
> > >
> > > The empirical evidence is consistent with this interpretation. The comparison with **AlphaRouter-Diff** (Appendix Table 4) shows that merely detecting whether the two models disagree under the same prefix is not sufficient, which reflects the limitation of a pure node-type / supervised-style route. The **AlphaRouter-GRPO** ablation in our rebuttal (see our response to **Reviewer Qtkw, W5 & Q4**) further shows that replacing the tree-advantage mechanism with a more standard rollout-based RL objective consistently degrades performance. Taken together, these results suggest that the gain of AlphaRouter arises specifically from converting trajectory returns into tree-structured node values and then using those values for localized credit assignment at divergent nodes.
> > >
> > > We hope this clarification helps address your remaining concerns.

---

### Official Review · Reviewer_vNzc · 2026-03-13

**Soundness:** 2
**Presentation:** 3
**Significance:** 3
**Originality:** 3
**Overall Recommendation:** 4
**Confidence:** 2

**Summary:**

The paper presents a method for collaborative computing using an SLM and an LLM where instead of trying to match the performance of the LLM, the authors identify that some routing paths can go above the LLM performance. To avoid the impractical search through all paths, they use Reinforcement Learning with a reward function based on contrasting the expected outcomes of each subpath to avoid relying on a sparse reward that is based on the final answer only.

**Compliance With Llm Reviewing Policy:**

Affirmed.

**Final Justification:**

I think the paper is useful and the authors addressed my concerns in the rebuttal. I'm maintaining my already positive score.

**Key Questions For Authors:**

- Why did you choose the current baselines and why did you decide not to includes ones like Co-LLM and CITER which you discuss in the related work?
- Taking into consideration that switching between the SLM and LLM doesn’t come for free on the systems side because the LLM needs to perform prefill on all tokens since its last generated token whenever triggered, does your routing actually provide real-world efficiency gains? Can you provide evaluations that show the actual benefit rather than AAP?

**Limitations:**

yes

**Strengths And Weaknesses:**

Strengths:

- Novel Contribution. I found the idea of exploring the various possible routing paths efficiently very interesting. Using recursive estimation of expected future reward and TAPO updates are clever ways to achieve that.
- Impressive Results against the baselines. The accuracy is higher than all the baselines and is getting closer to the LLM accuracy. the 0.5B+3B accuracy that exceeds the 3B accuracy is a good win for your unintuitive bet that you can use that collaboration to actually have better results than the LLM.

Weaknesses:

- Method of evaluating computational cost (AAP) seems to ignore the fact that using LLM to generate one token requires prefilling on all previously generated tokens and thus does not act as a proxy for real-world performance. Let's say you make the decision to route from the SLM to the LLM at token N. The LLM does not have the KV Cache values for all the previous tokens and thus it will have to run a prefill run on the LLM for all of unseen tokens rather than just a decode step for the new token generation. This fact seems to be completely ignored in the evaluation.

---

> ### Author Rebuttal · Authors · 2026-03-31
>
> We thank the reviewer for the positive comments and constructive questions.
>
> **Q1:** We do not include Co-LLM and CITER because a key practical advantage of our method is that it requires **no fine-tuning of either the SLM or the LLM**. Our router is trained on top of fixed backbone models, making the method simpler and more deployment-friendly. By contrast, both Co-LLM and CITER rely on additional **SLM adaptation/fine-tuning** in their released pipelines: Co-LLM includes base-model adaptation during deferral training, and CITER first fits the model on ground-truth reasoning data and then uses the adapted model to construct token-level labels/preferences for router training. Since these methods require extra model adaptation beyond routing itself, we do not consider them the most appropriate baselines for highlighting this practical advantage. Instead, we use R2R as the most directly comparable token-level routing baseline without such additional fine-tuning.
>
> **W1 & Q2:** We agree that AAP alone is insufficient to characterize real deployment efficiency, especially because invoking the LLM incurs non-trivial prefilling overhead. We have therefore added **actual end-to-end inference measurements**. All results are obtained on a single NVIDIA A800 80GB GPU, by randomly sampling 100 GSM8K problems and reporting average per-request statistics. We measure not only wall-clock latency and throughput, but also total computation (TFLOPs), total memory access (TB), and average memory access per token (GB/token), since LLM decoding is largely memory-bound.
>
> | Model       | CR.    | Param. (B) | Avg Tokens | Avg Latency (s) | Avg TPS | Acc. | TFLOPs | Total Memory Access (TB) | Mem/Token (GB) |
> | ----------- | ------ | ---------- | ---------- | --------------- | ------- | ---- | ------ | ------------------------ | -------------- |
> | Qwen-0.5B   | -      | 0.49       | 330.29     | 0.6789          | 485.35  | 45%  | 32.37  | 43.57                    | 1.319          |
> | Qwen-1.5B   | -      | 1.54       | 341.83     | 2.0207          | 169.38  | 73%  | 120.15 | 139.62                   | 4.085          |
> | Qwen-7B     | -      | 7.62       | 338.01     | 3.8420          | 88.00   | 91%  | 588.19 | 682.10                   | 20.180         |
> | SD          | -      | 8.10       | 326.95     | 2.5886          | 132.82  | 91%  | 605.64 | 708.46                   | 21.669         |
> | AlphaRouter | 8.74%  | 1.07       | 321.98     | 2.0821          | 165.14  | 78%  | 212.45 | 253.21                   | 7.864          |
> | AlphaRouter | 14.95% | 1.47       | 316.53     | 2.5582          | 134.46  | 86%  | 237.25 | 281.97                   | 8.908          |
> | AlphaRouter | 18.7%  | 1.72       | 328.97     | 2.9459          | 115.06  | 91%  | 259.92 | 305.06                   | 9.375          |
>
> These are wall-clock measurements and therefore already include practical switching overhead, including LLM prefilling cost. At the same 91% accuracy as Qwen-7B, AlphaRouter achieves lower latency (**2.9459s vs. 3.8420s**) and higher throughput (**115.06 vs. 88.00 tokens/s**), while substantially reducing computation and memory traffic (**259.92 vs. 588.19 TFLOPs**, **305.06 vs. 682.10 TB** total memory access).
>
> These results also clarify the comparison with **speculative decoding (SD)**. SD achieves strong latency at the high-accuracy point (**2.5886s**, **132.82 TPS**, **91%**), but with higher computation and memory traffic than standalone Qwen-7B (**605.64 vs. 588.19 TFLOPs**, **708.46 vs. 682.10 TB** memory access), reflecting the extra overhead of draft-and-verification. By contrast, AlphaRouter offers a more flexible trade-off: while slightly slower than SD at the matched 91% point, it can operate at much lower-cost regimes with only moderate accuracy sacrifice, e.g., **86%** accuracy at **2.5582s** and **134.46 TPS**, or **78%** accuracy at **2.0821s** and **165.14 TPS**. This kind of **controllable accuracy–efficiency frontier** is not naturally available to speculative decoding, whose goal is primarily to preserve the target model output.
>
> Because decoding is largely memory-bound, memory traffic is especially important for practical throughput. In this respect, AlphaRouter is highly efficient: at the 91% operating point, it reduces total memory access from **682.10 TB** to **305.06 TB**, and average memory access per token from **20.180 GB** to **9.375 GB**. Thus, AlphaRouter’s speedup comes not only from activating fewer parameters, but also from substantially alleviating the memory-access bottleneck in autoregressive decoding.
>
> We thank the reviewer again for taking the time to read our rebuttal, and we hope the above clarifications help resolve your concerns regarding baseline choice and practical efficiency.

---

> > ### Author Rebuttal · Reviewer_vNzc · 2026-04-04
> >
> > Thank you for addressing my concern. The new results show that SD is 1.2X slower than your fastest configuration which has a 13% Acc gap. I don't think this gives a big motivation for your method. What's the main benefit of the method? Also, is it possible to list the maximum memory used which indicates the size of the GPU memory needed rather than memory transfer which is somewhat included in the latency numbers?
> > So, I'm keeping my already positive score.

---

> > > ### Author Response · Authors · 2026-04-07
> > >
> > > Thank you for the helpful follow-up and for keeping your positive score. We agree that the main benefit of AlphaRouter should be stated more clearly. Our goal is **not** to claim that AlphaRouter universally dominates speculative decoding at a matched-accuracy point. Rather, the main benefit is that it provides a **controllable accuracy–efficiency frontier** through selective LLM usage. In other words, AlphaRouter is designed for **budget-aware collaborative inference**, where one may deliberately trade a small or moderate amount of accuracy for substantially lower compute and memory cost, depending on deployment constraints.
> > >
> > > This is also why we view the comparison with SD as complementary rather than purely head-to-head. Speculative decoding is primarily designed to **preserve the target model’s output** while accelerating generation, so it is naturally strongest when the deployment objective is to stay as close as possible to the target-model accuracy. By contrast, AlphaRouter is designed to **adaptively allocate expensive LLM computation only where it is most beneficial**, thereby exposing multiple operating points under different latency/cost budgets. Therefore, we believe the main value of the method lies in the **flexible Pareto frontier**, rather than only in whether the single fastest configuration beats SD at a matched accuracy point.
> > >
> > > We also agree that **peak VRAM** is an important deployment metric. Please see our response to Reviewer 9tUx for the detailed peak-VRAM results. This also helps clarify the reviewer’s concern about peak VRAM. At deployment, AlphaRouter keeps both the 0.5B and 7B models resident and maintains KV caches for both. However, in our profiling, the additional footprint from the 0.5B side is small compared with the 7B model itself: at test time, the 7B model uses **14.18 GiB** for weights and **0.04 GiB** for KV cache, while the 0.5B model uses only **0.92 GiB** for weights and **0.01 GiB** for KV cache. Therefore, the deployment footprint remains close to the standalone 7B case. Relative to a standalone 7B baseline (about **14.22 GiB** from 7B weights + 7B KV), AlphaRouter uses about **15.37 GiB** of attributable test-time memory, i.e., an increase of only **1.15 GiB**. For long CoT decoding, this overhead is mainly due to the additional small model and router, rather than test-time tree materialization or unusually large KV-cache growth. This is also broadly consistent with other **two-model collaborative inference methods**, such as speculative decoding, which likewise keep both a small and a large model resident during decoding.

---

### Decision · Program_Chairs · 2026-04-30

**Decision:**

Accept (regular)

**Comment:**

All reviewers found the problem of exploring possible routing paths to optimize efficiency very novel and interesting, the approach of recursively estimating TAPO updates clever, and the experimental results convincing for justifying the effectiveness of the approach. A major concern among reviewers are the overhead and additional costs of the proposed methods, but the authors did a good job in addressing this concern during the rebuttal. While ultimately the paper remains borderline and some reviewers still had reservation about the cross-task or cross-domain generalization, the review team overall feels that the merits of the paper outweigh this potential limitation.

Overall, the review team agrees that the paper could be accepted to the conference if there is room.